# LogicSAGE: Neuro-Symbolic Reasoning with Socratic-Guided Enhancement

**Jinlong Tian** [* 1]  **Jiang Yu** [* 1]  **Kewei Cheng** [2]  **Fengxiang Cheng** [3]  **Yue He** [4]  **Yunfei Wang** [1]  **Haotian Wang** [1]  **Haoxuan Li** [5]  **Wenjing Yang** [1]  **Shixuan Liu** [1]

## Abstract

Large Language Models (LLMs) often struggle with complex logical reasoning. Existing approaches typically rely on either purely neural reasoning in natural language or offloading to formal solvers via symbolic representations. However, both paradigms face significant limitations: while LLMs exhibit strong semantic intuition they are prone to hallucinations, whereas symbolic solvers offer rigorous derivation but remain highly sensitive to minor syntactic errors. To combine the strengths of these two paradigms while mitigating their respective limitations, we introduce **LogicSAGE** (**L**ogic-informed **S**ocratic **A**gent for **G**uided **E**nhancement), a dual-process framework that integrates a robust neural reasoner (System 1) with a rigorous symbolic validator (System 2). Specifically, our framework employs a Socratic Error Correction mechanism that treats solver feedback not as terminal failures but as pedagogical signals, engaging in a dialectic loop to iteratively refine logic programs and resolve semantic ambiguities. Extensive experiments on five benchmarks show that LogicSAGE (8B) achieves a state-of-the-art 92.36% average accuracy, significantly outperforming GPT-4 baselines, which establishes that architectural innovation can supersede model scale in faithful reasoning.

## 1. Introduction

Equipping Large Language Models (LLMs) with robust reasoning capabilities is essential for advanced AI, yet their current proficiency remains limited. On one hand, LLMs

have achieved mastery over linguistic fluency and semantic intuition, exhibiting emergent capabilities akin to Kahneman's *System 1* fast thinking (Wei et al., 2023; Kahneman, 2011). However, they fundamentally rely on probabilistic correlation rather than axiomatic derivation, rendering them prone to hallucinations (Dziri et al., 2023) and logical inconsistencies, especially in multi-step deduction (Ji et al., 2023; Wei et al., 2022). On the other hand, symbolic systems (e.g., Theorem Provers, SAT Solvers) offer the precision and explainability of *System 2* slow thinking but lack the flexibility to parse the ambiguity of natural language.

Neuro-Symbolic (NeSy) AI has emerged as a promising paradigm (Pan et al., 2023), typically employing LLMs to translate natural language into formal logic programs (e.g., FOL, Python) for deterministic execution (Olausson et al., 2023; Cheng et al., 2023; Liu et al., 2024). However, as illustrated in Figure 1(a), existing pipelines exhibit a spectrum of robustness strategies. Early approaches adopt a *Generate-and-Hope* strategy, relying on brute-force over-generation with majority voting. More recent methods such as Logic-LM (Pan et al., 2023) incorporate reactive self-correction based on solver error messages, and Logic-LM++ (Kirtania et al., 2024) extends this with multi-step refinement. Despite these advances, current pipelines share two persistent bottlenecks:

- Translation Fragility: symbolic solvers are highly sensitive to syntactic misalignment (Liu et al., 2025), where even minor discrepancies cause execution failure.

- Silent Failure: since solvers faithfully execute whatever formula they receive, a semantically incorrect LLM translation (e.g., generating $\wedge$ instead of $\neg A \wedge \neg B$ for "Neither A nor B") produces a valid execution with an unsound conclusion—and existing reactive correction mechanisms, which rely on explicit solver errors, provide no signal for such cases (Parthasarathy et al., 2024).

Consequently, even with iterative refinement, existing pipelines remain fundamentally *reactive*: they can repair programs that crash, but lack the means to detect or correct semantically flawed programs that execute successfully.

To address these issues, we synergize the semantic flexibility of LLMs with the deductive rigor of symbolic solvers,

---

*Equal contribution [1] College of Computer Science and Technology, National University of Defense Technology, Hunan, China [2] Microsoft, USA [3] Tsinghua University, Beijing, China [4] School of Information, Renmin University of China, Beijing, China [5] Peking University, Beijing, China. Correspondence to: Shixuan Liu <szftandy@hotmail.com>.

*Proceedings of the 43rd International Conference on Machine Learning*, Seoul, South Korea. PMLR 306, 2026. Copyright 2026 by the author(s).

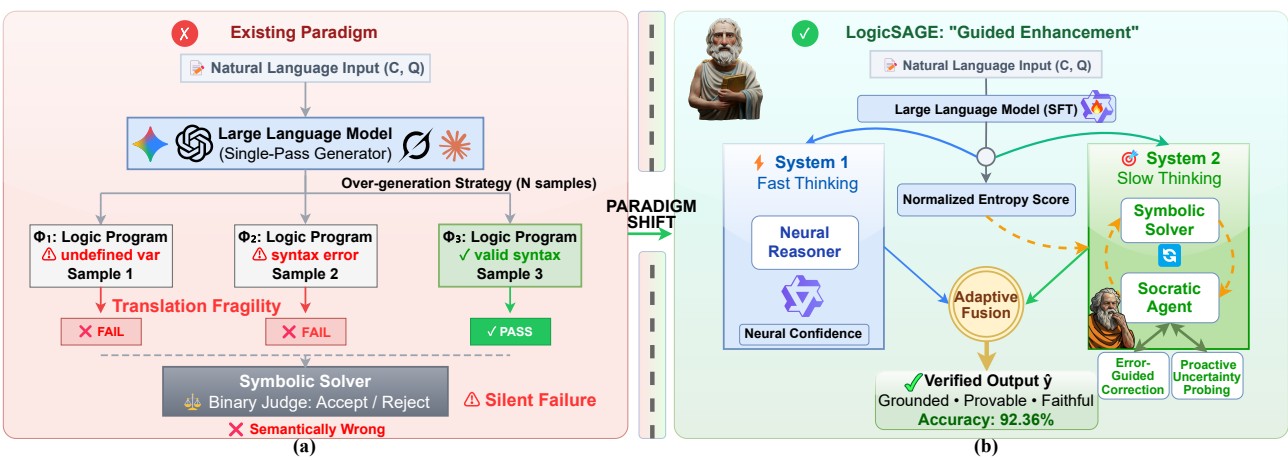

*Figure 1.* The Paradigm Shift in Neuro-Symbolic Reasoning. (a) Existing reactive pipelines suffer from *Translation Fragility*, where solver errors either cause terminal failure or, when execution succeeds, leave semantic misalignments undetected. (b) **LogicSAGE** introduces *Guided Enhancement* via a Dual-Process architecture. It features a *Socratic Error Correction* loop that iteratively repairs logic programs based on solver feedback, synergizing with a neural reasoner for robust execution.

introducing LogicSAGE (Logic-informed Socratic Agent for Guided Enhancement). Beyond reactive error correction, our framework introduces proactive semantic verification, upgrading the neuro-symbolic pipeline from purely reactive repair to guided, diagnostically-informed enhancement. As illustrated in Figure 1, LogicSAGE implements a coherent dual-process architecture that integrates a *System 1* neural reasoner with a *System 2* symbolic validator. At its core is a Socratic Agent that orchestrates a dialectic reasoning loop simulating a structured teacher-student interaction through two complementary mechanisms: (1) *Reactive Error-Guided Correction* turns the solver's intolerance into a source of supervision: rather than discarding programs that raise syntax errors, the agent treats precise diagnostic signals as pedagogical feedback, guiding iterative, targeted debugging instead of blind regeneration(Weng et al., 2023; Chen et al., 2023; Shinn et al., 2023). This reactive loop constitutes the primary driver of performance gains by resolving the dominant mode of failure. (2) *Proactive Uncertainty Probing* further employs an entropy-aware verification protocol to detect latent semantic ambiguities, such as conflating logical operators, even when code compiles successfully. By monitoring token-level entropy spikes, the system proactively interrogates uncertain logical connectives, ensuring programs are both syntactically valid and semantically grounded. Finally, to guarantee robustness, we design an *Adaptive Fusion* mechanism that dynamically prioritizes rigorous symbolic execution while leveraging neural intuition as a fail-safe when formalization is intractable.

The contributions of this work are three-fold:

- Reflexive NeSy Architecture: We propose LogicSAGE, which integrates a dynamic Socratic debugging loop into the NeSy pipeline. This mechanism substantially miti-

gates the inherent translation fragility bottleneck, thereby transforming execution errors into constructive supervision signals.

- Entropy-Aware Proactive Probing: We introduce a novel mechanism to detect silent failures. By monitoring token-level normalized entropy streams, our agent identifies and corrects latent semantic hallucinations even when symbolic verification succeeds.

- Comprehensive Evaluation: Experiments on five benchmarks show that LogicSAGE (8B) achieves a new SOTA average accuracy of 92.36%, significantly outperforming GPT-4 baselines and establishing that specialized architectural innovation can bridge the performance gap between parameter-efficient models and large-scale giants.

## 2. Related Work

### 2.1. Prompting Strategies for Logical Reasoning

The emergence of Chain-of-Thought (CoT) prompting (Wei et al., 2022) marked a significant milestone, enabling LLMs to perform multi-step deduction by generating intermediate reasoning traces. Several extensions have since been proposed to enhance logical consistency (Wang et al., 2022). Logic-of-Thought (LoT) (Liu et al., 2025) injects propositional logic derived from context into prompts to ground the reasoning process. Similarly, Tree of Thoughts (ToT) (Yao et al., 2023) and Graph of Thoughts (Besta et al., 2024) explore non-linear reasoning paths. However, despite these structural enhancements, purely prompt-based methods remain fundamentally bound by the probabilistic nature of LLMs. They lack a rigorous verification mechanism, leaving them prone to hallucinations and unfaithful reasoning, particularly in complex proofs where a single false step

invalidates the entire chain (Maynez et al., 2020).

## 2.2. Neuro-Symbolic Reasoning Frameworks

To overcome the unreliability of pure LLMs, Neuro-Symbolic (NeSy) AI delegates the reasoning burden to deterministic solvers. Pipeline Approaches. A dominant paradigm involves using LLMs as semantic parsers to translate Natural Language (NL) into formal specifications (e.g., FOL, Python), which are then executed by external solvers. LINC (Olausson et al., 2023) and NL2FOL (Lalwani et al., 2024) demonstrate that offloading deduction to theorem provers (e.g., Prover9) significantly boosts performance on benchmarks like FOLIO and ProofWriter. Logic-LM (Pan et al., 2023) further refines this by integrating a self-correction mechanism based on solver error messages. More recently, Logic-LM++ (Kirtania et al., 2024) extends this with multi-step refinement and paired comparisons, while LTRAG (Hu et al., 2025) enhances the process via retrieval-augmented generation of similar logical templates. VERUS-LM (Callewaert et al., 2025) and Aristotle (Xu et al., 2025) propose modular architectures that decompose complex problems into sub-tasks solved by specialized symbolic engines. Nevertheless, these pipelines remain vulnerable to translation fragility: most refinement mechanisms prioritize syntactic validity over semantic faithfulness, often leading to *semantic misalignment* (silent failures) where executable code misinterprets premises. Furthermore, the rigid dependency on solvers frequently results in low recall when formalization fails, a limitation LogicSAGE addresses through its adaptive dual-process architecture.

## 2.3. Logic-Enhanced Fine-Tuning and Optimization

Instead of relying solely on in-context learning, recent works explore fine-tuning LLMs to better understand logic. LogicLlama (Yang et al., 2024) and Symbol-LLM (Xu et al., 2024a) employ Supervised Fine-Tuning (SFT) on large-scale NL-FOL pairs to improve translation accuracy. LOGICPO (Viswanadha et al., 2025) utilizes Direct Preference Optimization (DPO) to align LLMs with valid logical forms, reducing syntax errors. ProverGen (Qi et al., 2025) and Logic-Thinker (Wen et al., 2025) construct synthetic datasets with solver-verified reasoning chains to teach LLMs to think like a prover. Although fine-tuning significantly bolsters base capabilities, it does not fully eliminate inherent stochasticity, leaving models prone to hallucination under distribution shift without runtime safeguards(Madaan et al., 2023; **?**; **?**). LogicSAGE distinguishes itself by targeting this inference-time gap: rather than expensive retraining, it introduces a training-free Socratic loop(Du et al., 2023) to dynamically verify and refine logic programs, ensuring robustness through adaptive neuro-symbolic fusion.

## 3. Problem Formulation

We consider a logical reasoning task defined by a dataset $\mathcal{D} = \{(x_i, y_i)\}_{i=1}^N$, where input $x = (C, Q)$ comprises a natural language context and query, and $y \in \mathcal{Y}$ denotes the ground-truth label. To bridge the gap between informal language and formal logic, we adopt a neuro-symbolic decomposition where the inference function is composed of two distinct modules: a neural semantic parser $\mathcal{G}_\theta$ (parameterized by an LLM) that maps the input $x$ to an intermediate logic program $\Phi = \mathcal{G}_\theta(x)$, and a deterministic symbolic solver $\mathcal{E}$ that executes $\Phi$ to derive the final answer $y = \mathcal{E}(\Phi)$. Consequently, our objective is to ensure that the generated program $\Phi$ is both syntactically valid and semantically faithful, such that its execution result matches the ground truth, i.e., $\mathcal{E}(\Phi) = y$.

## 4. Method

In this section, we present LogicSAGE, a unified neuro-symbolic framework inspired by the Dual-Process Theory in cognitive science. As illustrated in Figure 2, our architecture synergizes a *System 1* intuitive reasoner (Neural CoT) with a *System 2* analytical engine (Symbolic Solver) through a comprehensive five-stage pipeline (Input, Parse, Reason, Fuse, Output). The workflow begins with an Entropy-Aware Semantic Parser that translates natural language into logic programs while monitoring internal uncertainty. This is mediated by a novel Socratic Agent, which orchestrates both reactive debugging (for syntax errors) and proactive probing (for silent failures), before an Adaptive Fusion Mechanism dynamically integrates the neural and symbolic outputs to ensure both soundness and coverage.

### 4.1. Neural Semantic Parsing

The first module serves as the interface between natural language and formal logic.

#### 4.1.1. LOGIC PROGRAM GENERATION

We employ a fine-tuned LLM as the generator $G_\theta$. Given input $(C, Q)$, the model generates a logic program $\Phi$ via $\Phi = G_\theta(C, Q, \mathcal{T}_{schema})$, where $\mathcal{T}_{schema}$ represents the domain-specific schema. As depicted in the *PARSE* module of Figure 2, our system adaptively maps different reasoning types to optimal formalisms: Deductive Reasoning is mapped to Logic Programming (LP), First-Order Logic tasks to FOL, Constraint Satisfaction problems to CSP, and Analytical Reasoning tasks to SAT solvers.

#### 4.1.2. ENTROPY-AWARE UNCERTAINTY

Detecting silent failures requires a metric that reflects the model's internal certainty. However, using raw entropy

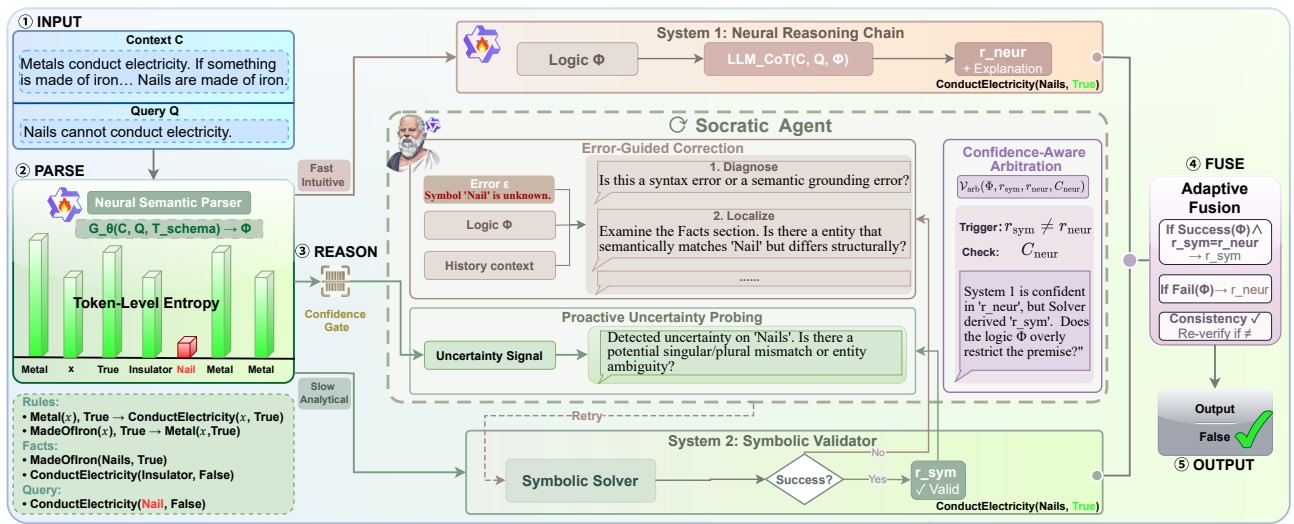

*Figure 2.* Overview of the LogicSAGE Framework. The pipeline consists of five stages: ① Input encoding; ② Entropy-Aware Semantic Parsing, which translates natural language $(C, Q)$ into logic programs $(\Phi)$ while monitoring token-level entropy to detect potential hallucinations; ③ Dual-Process Reasoning, featuring a *System 1* Neural Reasoning Chain and a *System 2* Symbolic Validator. The interaction is orchestrated by a *Socratic Agent* with three distinct mechanisms: *Error-Guided Correction* for syntax repairs (using error $\epsilon$), *Proactive Uncertainty Probing* for silent semantic failures (using entropy signals), and *Confidence-Aware Arbitration* to resolve conflicts between symbolic derivation $(r_{sym})$ and neural intuition $(r_{neur})$; ④ Adaptive Fusion, which selects the final prediction $\hat{y}$ via a hierarchical gating strategy; and ⑤ Final Output generation.

directly is flawed due to the distributional shift between distinct cognitive processes. Structural tokens (syntax) typically exhibit much lower entropy than grounding tokens (semantics). Merging them into a single metric obscures specific semantic hallucinations. To address this, we introduce an Entropy-Aware Uncertainty Normalization framework.

**Stream Separation.** We first define a partition function that segregates the generated token sequence $\Phi$ into two distinct streams: the syntactic stream $K_{syn}$ (closed-vocabulary operators, e.g., $\forall, \wedge$) and the semantic stream $K_{sem}$ (open-vocabulary predicates, constants). This separation addresses the fundamental distributional disparity between streams, ensuring semantic tokens are normalized against their own statistical baseline rather than being masked by low-entropy syntax.

**Normalized Entropy Score (NES).** To make entropy values comparable and robust to the inherent difficulty of the problem, we perform intra-stream normalization. We compute the Z-score of each token's entropy relative to the statistics of its own stream. For a stream $S \in \{syn, sem\}$, let $\mu_S$ and $\sigma_S$ be the mean and standard deviation of entropy within that stream. The Normalized Entropy Score (NES), $Z_S(t)$, is defined as:

$$Z_S(t) = \frac{H(t) - \mu_S}{\sigma_S + \epsilon}, \quad \forall t \in K_S \qquad (1)$$

By normalizing against the local baseline $\mu_S$, this metric effectively filters out the background noise of syntactic confidence, highlighting relative anomalies.

**Spike-Based Detection.** We identify potential failures by detecting Entropy Spikes—tokens that exhibit statistically significant deviations. A spike is flagged if:

$$\mathcal{T}_{spike}^S = \{t \in K_S \mid Z_S(t) > \alpha\} \qquad (2)$$

where $\alpha$ is typically set to 3. The Socratic Agent is triggered if and only if $|\mathcal{T}_{spike}^S| > 0$. Crucially, this set formulation enables the agent to perform *targeted verification*, focusing its dialectic inquiry solely on the specific elements $t \in \mathcal{T}_{spike}^S$ (e.g., a specific predicate *MadeOf*) rather than re-evaluating the entire logical formula.

### 4.1.3. SEMANTIC ALIGNMENT VIA BACK TRANSLATION

To mitigate the translation gap where generated formulas are syntactically correct but semantically divergent, we introduce a verification step. We employ a separate LLM instance to back-translate $\Phi$ into natural language $C'_{rec}$ and compute a semantic consistency score $S_{sem}$ as $S_{sem} = \text{Sim}(C, C'_{rec})$, If $S_{sem} < \tau$ (a predefined threshold), the generation is flagged for regeneration, ensuring the logic program faithfully represents the original premises.

### 4.2. System 1: Neural Reasoning Chain

Representing the fast thinking component of our dual-process architecture, this module serves as a robust engine for scenarios where strict formalization is incomplete or intractable (e.g., questions requiring external commonsense). Here, we utilize the LLM to perform symbolic-augmented

CoT, yielding $\hat{y}_{neur}, \text{Expl} = \text{LLM}_{CoT}(C, Q, \Phi)$. Unlike standard CoT, we explicitly inject the generated logic program $\Phi$ back into the prompt as a structured context. This forces the neural reasoner to ground its intuitive deductions in the formal skeleton it previously generated, thereby reducing hallucinations while leveraging its inherent flexibility to interpret ambiguous predicates that might otherwise crash a deterministic solver.

### 4.2.1. NEURAL CONFIDENCE ESTIMATION

To quantify the reliability of the System 1 reasoning chain, simply averaging token probabilities is insufficient, as it allows high-confidence "filler" tokens to mask localized hallucinations. Instead, we adopt a *weakest-link* formulation. Given a generated reasoning chain $R = (t_1, \ldots, t_T)$, we calculate the Neural Confidence $C_{neur}$ by identifying the minimum confidence over a sliding window of size $W$ with stride $S$. To mitigate synonym variance, we utilize a smoothed token confidence $c_t$ based on the top-$K$ candidates:

$$C_{neur} = \min_i \left( \frac{1}{W \cdot K} \sum_{j=0}^{W-1} \sum_{k \in \text{TopK}(t_{i \cdot S + j})} p_\theta(x_k \mid x_{<(i \cdot S + j)}) \right) \tag{3}$$

This metric ensures that the most fragile segment in the deduction determines the overall confidence, aligning with the strict nature of logical proofs.

### 4.3. System 2: Symbolic Validator with Dual-Mode Socratic Agent

Parallel to the neural pathway, this module embodies the deliberative "slow thinking" analytical process, strictly executing $\Phi$ via a deterministic solver $E$. However, symbolic systems face two hurdles: *Translation Fragility* and *Silent Failure*. To address both, our Socratic Agent operates in two modes, simulating a structured teacher-student dialectic. We instantiate 'Teacher' and 'Student' as distinct personas within a single LLM via prompt engineering. The Teacher analyzes diagnostic signals to pose pedagogical questions, guiding the Student's iterative code refinement. The complete execution protocol is formalized in Algorithm 1.

### 4.3.1. MODE 1: REACTIVE ERROR-CORRECTION.

This mode activates when the solver $E(\Phi)$ returns a runtime error $\epsilon$ (e.g., `SyntaxError`). Unlike standard self-correction methods that simply feed the error back, we establish a multi-turn interaction. The system constructs a diagnostic prompt $\mathcal{P}_{refine} = (\Phi, \epsilon, \mathcal{I}_{history})$. Guided by this context, the Socratic Teacher Agent poses a series of pedagogical questions, leading the Student Agent to engage in reflexive self-correction. This cycle continues iteratively until the solver returns a valid result or the maximum depth $T_{max}$ is reached, effectively transforming opaque error signals into constructive reasoning chains.

---

**Algorithm 1** Dual-Mode Socratic Reasoning Loop

**Require:** Context $C$, Query $Q$, Depth $T_{max}$, Threshold $\alpha$
1: $\Phi \leftarrow \text{Generator}(C, Q)$
2: **for** $t = 1$ to $T_{max}$ **do**
3:     $r_{sym}, \text{status} \leftarrow \text{Solver}(\Phi)$
4:     **if** status == Error **then**
5:        {Mode 1: Reactive}
6:        $\Phi \leftarrow \text{Student}(\Phi, textTeacher_{\text{React}}(\Phi, \text{Trace}(r_{sym})))$
7:     **else if** $\max(\text{CalcEntropy}(\Phi)) > \alpha$ **then**
8:        {Mode 2: Proactive}
9:        $\mathcal{T}_{spike} \leftarrow \{tok \mid \text{NES}(tok) > \alpha\}$
10:      $\Phi \leftarrow \text{Student}(\Phi, \text{Teacher}_{\text{Proact}}(\Phi, \mathcal{T}_{spike}))$
11:     **else**
12:        **break** {Valid & Grounded}
13:     **end if**
14: **end for**
15: **return** $\Phi, r_{sym}$

---

### 4.3.2. MODE 2: PROACTIVE UNCERTAINTY PROBING.

Crucially, successful execution ($S(\Phi) = \text{Success}$) does not guarantee semantic correctness. For instance, a model might incorrectly translate the linguistic negative disjunction "*Neither A nor B*" into a logical conjunction ($A \wedge B$). While syntactically valid in FOL, this completely reverses the intended meaning (see **Case 1 in Appendix A**). To detect "Silent Failures", this mode activates when the solver returns a result. The agent scrutinizes the Normalized Entropy Scores (NES) defined in Sec. 4.1.2. If a *Semantic Spike* ($\mathcal{S}_{spike}^{sem}$) is detected, the Socratic Teacher Agent initiates a verification inquiry. The Student Agent must then justify or revise the mapping, ensuring the final proof is not only executable but also grounded.

### 4.4. Adaptive Result Fusion

The final decision $\hat{y}$ is derived via a hierarchical gating mechanism to balance the provable soundness of *System 2* with the robust coverage of *System 1*. Let $r_{sym}$ denote the deterministic result from the Symbolic Validator and $r_{neur}$ denote the probabilistic prediction from the Neural Reasoning Chain. To synthesize these distinct modalities, we formulate the fusion strategy as a priority cascade.

### 4.4.1. LOGIC-FIRST, INTUITION-FALLBACK.

The primary gate is the execution status $\mathcal{S}(\Phi)$ of the logic program. *System 2* is prioritized for its rigorous derivation. However, unlike prior works that discard failed executions, we treat *System 1* as a critical fail-safe mechanism. Even if formalization proves intractable (System 2 fails), the neural intuition of System 1 remains a valuable signal, ensuring the system always provides a best-effort response rather than abstaining.

### 4.4.2. CONFIDENCE-AWARE SOCRATIC ARBITRATION.

A subtle challenge arises when System 2 executes successfully but contradicts System 1 (i.e., $r_{sym} \neq r_{neur}$). This discordance requires context-dependent resolution, as detailed in **Appendix A (Case 2)**. For strict deductive tasks where neural models often drift, the Arbiter prioritizes the rigorous symbolic derivation (Case 2.1). Conversely, for tasks involving linguistic ambiguity or implicit commonsense where symbolic formalization is brittle (high program entropy), the Arbiter favors the high-confidence neural intuition (Case 2.2). This module employs a Socratic Arbiter $\mathcal{V}_{arb}$ that weighs the rigour of the symbolic proof against the certainty of the neural intuition to determine the final output.

The fusion function is formally defined as:

$$\hat{y} = \begin{cases} r_{\text{sym}}, & \text{if } \mathcal{S}(\Phi) = \text{Success} \wedge r_{\text{sym}} = r_{\text{neur}}; \\ \mathcal{V}_{\text{arb}}(\Phi, r_{\text{sym}}, r_{\text{neur}}, C_{\text{neur}}), & \text{if } \mathcal{S}(\Phi) = \text{Success} \wedge r_{\text{sym}} \neq r_{\text{neur}}; \\ r_{\text{neur}}, & \text{otherwise (i.e., } \mathcal{S}(\Phi) = \text{Fail}). \end{cases} \tag{4}$$

where $\mathcal{S}(\cdot)$ represents the solver's execution status. In the conflict case (second row), the Socratic Arbiter $\mathcal{V}_{arb}$ reviews the translation $\Phi$ and the confidence $C_{neur}$. For instance, if $r_{sym}$ contradicts a highly confident $r_{neur}$ ($C_{neur} \gg$ threshold), the Agent is prompted to scrutinize $\Phi$ for semantic misalignments (e.g., misinterpretation of predicates); conversely, if $C_{neur}$ is low, the rigorous symbolic result $r_{sym}$ is preferred. This hybrid architecture ensures that LogicSAGE benefits from the "best of both worlds": the provable correctness of symbolic logic and the broad coverage of neural intuition.

## 5. Experiments

In this section, we provide a comprehensive empirical evaluation of LogicSAGE. We aim to answer the following research questions: **RQ1 (Performance)**: How does LogicSAGE compare against state-of-the-art LLMs and existing neuro-symbolic frameworks across diverse reasoning tasks? **RQ2 (Socratic Effectiveness)**: How do the *Reactive* (Error-Guided) and *Proactive* (Entropy-Guided) modes contribute individually to resolving "Translation Fragility" and "Silent Failures"? **RQ3 (Dual-Process Necessity)**: Is the Dual-Process Fusion mechanism essential? How does the synergy between *System 1* (Neural) and *System 2* (Symbolic) compare to relying on either modality alone?

### 5.1. Experimental Setup

We evaluate LogicSAGE on five logical reasoning benchmarks: PrOntoQA (Saparov & He, 2022), ProofWriter (Tafjord et al., 2021), FOLIO (Han et al., 2024), LogicalD-

eduction (Srivastava et al., 2023), and AR-LSAT (Zhong et al., 2022). We compare against baselines including GPT-4 (Standard/CoT) and neuro-symbolic frameworks like Logic-LM (Pan et al., 2023) and VERUS-LM (Callewaert et al., 2025). All our models utilize `Qwen-3-8B` as the backbone, we fine-tuned the model on a solver-verified dataset synthesized by the more capable `Qwen-3-Max`. Detailed dataset statistics, baseline descriptions, and implementation hyperparameters are provided in Appendix C.

### 5.2. Main Results

#### 5.2.1. OVERALL ACCURACY

Table 1 compares performance across five logical reasoning benchmarks. Despite utilizing a compact `Qwen-3-8B` backbone, our model establishes a new state-of-the-art average accuracy of 92.36%. To isolate architectural gains from backbone strength, we conduct controlled comparisons where all methods share the identical `Qwen-3-8B` backbone (Table 1, lower block). Under this setting, our training-free LogicSAGE (80.30%) already outperforms all controlled baselines including VERUS-LM (79.82%), and the gap widens substantially with SFT (92.36%). Logic-SAGE also generalizes across backbones, achieving 86.10% with `GPT-4`—surpassing all GPT-4-based baselines. These results confirm that the performance gains stem primarily from the Socratic neuro-symbolic architecture rather than the backbone alone. Furthermore, the substantial 12% gain over the base model underscores the transformative role of Supervised Fine-Tuning (SFT). Rather than merely adapting the style, SFT acts as a precision alignment step, effectively constraining the 8B model's probabilistic outputs to meet the strict syntactic rigor required by the symbolic solver.

#### 5.2.2. EXECUTABILITY AND RELIABILITY ANALYSIS

To investigate how LogicSAGE resolves the "Translation Fragility" bottleneck, we compare the Execution Rate (Er) (compilation success) and the Execution Accuracy (Ea) (percentage of valid programs matching ground truth). As shown in Table 2, LogicSAGE effectively solves the crash-proneness inherent in symbolic generation. This capability is most strikingly illustrated on the complex AR-LSAT benchmark, where GPT-4 based baselines suffer from catastrophic failure rates (Er < 40%). In contrast, our SFT model achieves an impressive Er of 95.6%. Crucially, LogicSAGE demonstrates superior reliability across different logic types. On four out of five benchmarks (PrOntoQA, ProofWriter, LogicalDeduction, and FOLIO), the model achieves a perfect 100% Er. This establishes that the Socratic Error Correction mechanism successfully eliminates syntax errors even in complex First-Order Logic tasks. Furthermore, on the challenging FOLIO dataset, LogicSAGE (SFT) demonstrates superior semantic fidelity compared to the advanced

*Table 1.* Accuracy (%) comparison on five logical reasoning benchmarks. Best results are bolded; best baseline results are underlined. The upper block reports results from original papers with their native backbones; the lower block presents controlled comparisons using the same Qwen3-8B backbone to isolate architectural contributions.

| Method | PrOntoQA | ProofWriter | FOLIO | LogicalDed. | AR-LSAT | Avg. |
|---|---|---|---|---|---|---|
| *Pure Neural Prompting Baselines (Original Backbones)* | | | | | | |
| GPT-4[†] (Standard) | 77.40 | 52.67 | 69.11 | 71.33 | 33.33 | 60.77 |
| GPT-4-CoT[†] | 98.79 | 68.11 | 70.58 | 75.25 | 35.06 | 69.56 |
| SymbCOT[*] | 99.60 | 82.50 | 83.33 | 93.00 | 43.91 | 80.47 |
| *Neuro-Symbolic Baselines (Original Backbones)* | | | | | | |
| Logic-LM (GPT-4)[*] | 83.20 | 78.80 | 68.55 | 87.63 | 23.40 | 68.32 |
| LOGIC-LM++ (GPT-4)[*] | 92.62 | 79.66 | 84.80 | 91.17 | 46.32 | 78.91 |
| LTRAG (GPT-4)[*] | 95.68 | 85.93 | 78.57 | 91.36 | 68.40 | 83.99 |
| VERUS-LM[*] | 95.80 | 93.83 | 78.43 | 88.67 | 68.36 | 85.02 |
| *Backbone-Controlled Comparisons (Qwen3-8B)* | | | | | | |
| Direct Answer | 69.60 | 51.33 | 62.25 | 44.33 | 20.35 | 49.57 |
| CoT | 98.00 | 70.00 | 64.71 | 46.67 | 21.65 | 60.21 |
| SymbCOT | 97.60 | 69.83 | 68.57 | 82.33 | 29.05 | 69.48 |
| Logic-LM | 80.40 | 65.33 | 59.51 | 78.00 | 19.91 | 60.63 |
| LOGIC-LM++ | 89.00 | 73.50 | 69.41 | 86.12 | 34.24 | 70.45 |
| LTRAG | 89.40 | 82.50 | 66.42 | 85.33 | 53.25 | 75.38 |
| VERUS-LM | 93.40 | 89.17 | 72.92 | 86.96 | 56.63 | 79.82 |
| *Ours* | | | | | | |
| LogicSAGE (Qwen3-8B) | 92.30 | 82.50 | 75.40 | 92.50 | 58.80 | 80.30 |
| LogicSAGE (GPT-4) | 99.20 | 83.84 | 82.26 | 97.40 | 67.80 | 86.10 |
| **LogicSAGE (SFT Qwen3-8B)** | **100.00** | **96.80** | **88.50** | **99.50** | **77.00** | **92.36** |

*Note:* [†] indicates results from (Xu et al., 2024b). [*] indicates results from original papers. All Qwen3-8B baselines use the non-thinking mode for fair comparison.

*Table 2.* Analysis of the **Symbolic Validator (System 2)** component. Execution Rate (**Er, %**), Execution Accuracy (**Ea, %**).

| Model | PrOntoQA | | ProofWriter | | FOLIO | | LogicalDed. | | AR-LSAT | |
|---|---|---|---|---|---|---|---|---|---|---|
| | Er | Ea | Er | Ea | Er | Ea | Er | Ea | Er | Ea |
| Logic-LM (GPT-4) | 100.0 | 83.2 | 99.0 | 79.6 | 85.8 | 79.9 | 100.0 | 87.6 | 39.8 | 58.8 |
| LOGIC-LM++ | — | — | 99.0 | 79.6 | 86.7 | 85.8 | — | — | 32.0 | 66.2 |
| VERUS-LM | 98.2 | 97.6 | 99.0 | 94.8 | 100.0 | 78.4 | 99.3 | 89.3 | **98.7** | 69.3 |
| LogicSAGE (Qwen3-8B) | 95.3 | 92.1 | 97.0 | 81.1 | 93.0 | 79.1 | 97.0 | 92.4 | 75.3 | 71.4 |
| **LogicSAGE (SFT Qwen3-8B)** | **100.0** | **100.0** | **100.0** | **95.3** | **100.0** | **87.1** | **100.0** | **99.0** | 95.6 | **74.6** |

*Note:* "—" indicates data not reported.

modular solver VERUS-LM. While both models achieve 100% executability, our model significantly outperforms VERUS-LM in Ea. This distinction validates that Logic-SAGE generates not merely *compilable* code, but *logically correct* proofs that faithfully capture the problem semantics.

### 5.3. The Efficiency-Performance Trade-off

To determine the optimal configuration for real-world deployment, we analyze the critical trade-off between reasoning rigor and computational cost by plotting the Average

Accuracy against Inference Latency across varying refinement depths ($T_{max}$).

As visualized in **Figure 3**, the system exhibits a steep performance ascent from 80.30% at the baseline ($T = 0$) to a peak of 92.36% at $T = 3$. This significant gain reflects the Socratic Agent's dual capability: resolving fatal syntax errors via reactive correction and, crucially, rectifying silent semantic failures detected via Entropy-Aware Proactive Probing. However, this accuracy gain eventually saturates, with $T = 4$ yielding negligible marginal improve-

*Table 3.* Ablation analysis of the Dual-Mode Socratic Agent.

| Configuration | PrOntoQA | | ProofWriter | | FOLIO | | LogicalDed. | | AR-LSAT | | Avg. | |
|---|---|---|---|---|---|---|---|---|---|---|---|---|
| | Er | Ea | Er | Ea | Er | Ea | Er | Ea | Er | Ea | Er | Ea |
| (A) w/o SEC (Baseline) | 78.5 | 83.2 | 72.4 | 72.5 | 58.6 | 59.1 | 84.2 | 80.3 | 31.5 | 55.4 | 65.0 | 70.1 |
| (B) + Reactive Only (Mode 1) | **100.0** | 100.0 | **100.0** | 93.5 | **100.0** | 84.0 | **100.0** | 98.0 | 95.1 | 66.5 | 99.0 | 88.4 |
| (C) + Proactive Probing (Full) | **100.0** | **100.0** | **100.0** | **95.3** | **100.0** | **87.1** | **100.0** | **99.0** | **95.6** | **74.6** | **99.1** | **91.2** |
| *Improv. Reactive (Δ B-A)* | +21.5 | +16.8 | +27.6 | +21.0 | +41.4 | +24.9 | +15.8 | +17.7 | +63.6 | +11.1 | +34.0 | +18.3 |
| *Improv. Proactive (Δ C-B)* | +0.0 | +0.0 | +0.0 | +1.8 | +0.0 | +3.1 | +0.0 | +1.0 | +0.5 | +8.1 | +0.1 | +2.8 |

*Table 4.* Ablation study on the necessity of the Dual-Process architecture.

| Configuration | PrOntoQA | ProofWriter | FOLIO | LogicalDed. | AR-LSAT | Avg. |
|---|---|---|---|---|---|---|
| System 1 Only (Neural CoT) | 88.50 (↓11.5%) | 86.20 (↓10.6%) | 79.40 (↓9.1%) | 86.30 (↓13.2%) | 60.50 (↓16.5%) | 80.18 (↓12.2%) |
| System 2 Only (Symbolic) | 100.00 (↓0.0%) | 95.30 (↓1.5%) | 87.10 (↓1.4%) | 99.00 (↓0.5%) | 71.32 (↓7.4%) | 90.54 (↓2.0%) |
| **Full Model (LogicSAGE)** | **100.00** | **96.80** | **88.50** | **99.50** | **77.00** | **92.36** |

*Note:* Percentages in red with ↓ arrows indicate performance degradation compared to the full LogicSAGE model.

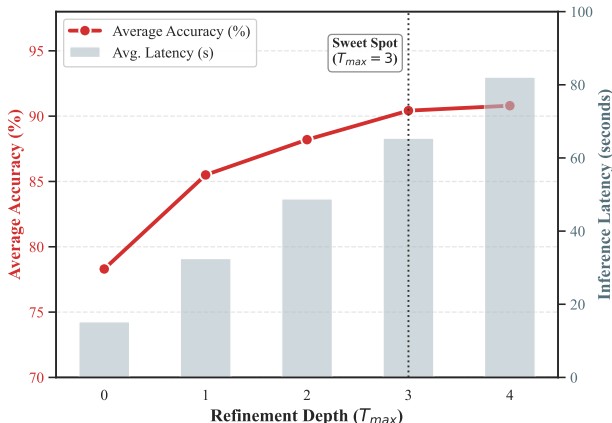

*Figure 3.* **Efficiency-Performance Trade-off.**

ments ($< 0.2\%$). Conversely, the computational cost follows a strict linear growth trajectory, adding approximately 15-20 seconds of latency per iteration due to the combined overhead of symbolic execution and LLM re-generation. With the total latency exceeding 80 seconds at $T = 4$, we identify $T_{max} = 3$ as the operational "sweet spot", striking the ideal balance that maximizes logical faithfulness within a practical time budget ($\sim 65$s).

## 5.4. Ablation Study

### 5.4.1. EFFECTIVENESS OF SEC

To rigorously quantify the contribution of the Socratic Error Correction (SEC) module and isolate its component-wise benefits. We decompose the SEC into two phases: *Reactive Correction* (syntax repair) and *Proactive Probing* (semantic verification). The results are detailed in Table 3.

**Solving Translation Fragility (Reactive Mode).** The base-

line's collapse on AR-LSAT (Row A) exposes the inherent Translation Fragility of compact models—they struggle to produce strictly valid code zero-shot. The introduction of *Reactive Error-Guided Correction* (Row B) fundamentally alters this dynamic. By treating solver error logs (e.g., `SyntaxError`) as pedagogical feedback rather than terminal failures, the model learns to iteratively debug its own output. The dramatic surge in execution rate on AR-LSAT (a +63.6 point leap) empirically validates this mechanism: it proves that iterative self-correction can effectively mitigate the strict dependency of syntactic competence on raw model scale, enabling an 8B model to attain the robustness typically reserved for significantly larger models.

**Resolving Silent Failures (Proactive Mode).** However, executability does not imply correctness. In Row (B), although the AR-LSAT execution rate is near-perfect (95.1%), the execution accuracy lags at 66.5%, indicating a significant presence of "Silent Failures"—where code compiles but is logically ungrounded. This is where *Proactive Uncertainty Probing* (Row C) proves critical. By leveraging the entropy-aware metric to detect and interrogate uncertain predicates, it significantly boosts faithfulness without requiring execution errors. On AR-LSAT, Proactive Probing improves Ea by a further +8.1 points, effectively closing the gap between syntactic validity and semantic correctness. This decomposition highlights that while Reactive correction ensures the *form* (syntax) is valid, Proactive probing ensures the *content* (grounding) is faithful.

### 5.4.2. DUAL-PROCESS SYNERGY

Table 4 empirically validates the architectural necessity of coupling neural intuition with symbolic rigor. The results reveal a clear dichotomy: while the *(*System 1 Only) baseline

offers flexibility, it hits a Probabilistic Ceiling, lagging behind the full model by over 12% due to a lack of axiomatic strictness. Conversely, although *(*System 2 Only) serves as the primary performance driver (90.54%), it encounters a "Formalization Barrier" in linguistically ambiguous scenarios—most notably on AR-LSAT, where it trails the full model by 7.4% due to intractable translation failures. The *Adaptive Fusion* mechanism effectively reconciles this trade-off by leveraging System 1 as a semantic fail-safe. By prioritizing symbolic derivation while dynamically falling back to neural reasoning for these specific edge cases, LogicSAGE salvages intractable queries to achieve the final 92.36% accuracy, proving that a hierarchical hybrid approach is essential to handle the long tail of reasoning tasks where logic is strict but language is fluid.

## 6. Conclusion

In this paper, we addressed the Translation Fragility bottleneck inherent in neuro-symbolic reasoning, where the intolerance of formal solvers to minor errors often cripples the potential of Large Language Models (LLMs). We introduced LogicSAGE, a dual-process architecture that harmonizes the linguistic intuition of neural networks with the axiomatic rigor of symbolic logic. By modeling the reasoning process not as a linear translation but as a reflexive dialogue, our framework employs a novel Socratic Error Correction mechanism. This approach advances beyond purely reactive error correction toward a *guided enhancement* loop, where solver feedback acts as a pedagogical signal to guide the LLM toward valid execution and semantically faithful reasoning. Empirical evaluations across multiple benchmarks demonstrate that LogicSAGE not only establishes a new state-of-the-art in accuracy but, more importantly, achieves high faithfulness by ensuring that answers are grounded in verifiable logical proofs rather than probabilistic hallucinations under diverse and challenging reasoning scenarios.

LogicSAGE demonstrates that introspective self-correction is pivotal for transparent, Grounded AI. Future work will extend the Socratic Agent to support diverse formalisms, such as Probabilistic Soft Logic, to capture real-world nuances. Additionally, to mitigate inference latency, we plan to integrate Reinforcement Learning to internalize pedagogical feedback directly into the model's weights. Finally, we aim to generalize this *Guided Enhancement* paradigm beyond logic puzzles to broader code generation and algorithmic reasoning tasks.

## Acknowledgments

This research is supported by the NUDT Youth Independent Innovation Science Fund under Grant No. ZK25-20.

## Impact Statement

This paper presents work whose goal is to advance the field of Machine Learning. There are many potential societal consequences of our work, none which we feel must be specifically highlighted here.

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

# A. Case Studies

In this section, we present concrete case studies to illustrate the efficacy of LogicSAGE's core mechanisms. **Case 1** demonstrates how *Proactive Uncertainty Probing* detects semantic misalignments that escape standard solvers. **Case 2** provides examples of the *Adaptive Fusion* mechanism, detailing how the Socratic Arbiter resolves conflicts between the Neural (System 1) and Symbolic (System 2) streams.

## A.1. Case 1: Intercepting Silent Failures via Entropy

We first consider a scenario involving the linguistic structure "*Neither A nor B is...*". This construct is a common source of translation errors, often being mapped to a logical conjunction rather than a conjunction of negations.

- **The Silent Failure (Phase 1):** The initial generation ($\Phi_0$) translates "Neither the wolf nor the fox is friendly" into `Friendly(Wolf) ∧ Friendly(Fox)`. Crucially, this formula is **syntactically valid** in First-Order Logic. A standard symbolic validator would execute it successfully, leading to a confidently wrong answer.

- **The Entropy Signal:** Although the model generated the token ($\wedge$), the **Entropy-Aware Semantic Parser** detected a statistically significant spike in the Normalized Entropy Score (NES) at this specific connective. This indicates that while the model selected the most probable token, it harbored underlying uncertainty regarding the mapping.

- **Socratic Intervention:** Triggered by this *Semantic Spike*, the Socratic Teacher Agent intervenes with a grounding question: *"You used a conjunction... Should this imply negation of both?"*

- **Refinement (Phase 2):** Guided by this probe, the Student Agent generates the correct formalism ($\Phi_1$): `¬Friendly(Wolf) ∧ ¬Friendly(Fox)`.

---

**Case 1: Proactive Probing for Semantic Consistency**

**Premise:** "Neither the wolf nor the fox is friendly."
**Initial Generation (Draft $\Phi_0$):**
`Friendly(Wolf) ∧ Friendly(Fox)`
*Status:* Syntactically Valid (Solver: Success). **BUT Logically Wrong (Silent Failure).**
*Signal:* High Entropy detected on token $\wedge$ (AND).
**Socratic Proactive Probe:**
"You used a conjunction ($\wedge$) with high uncertainty. The premise uses 'Neither...nor'. Should this imply negation of both?"
**Refined Generation ($\Phi_1$):**
`¬Friendly(Wolf) ∧ ¬Friendly(Fox)`
*Status:* Valid & Correct.

---

## A.2. Case 2: Adaptive Fusion and Socratic Arbitration

A unique feature of LogicSAGE is its ability to arbitrate between *System 1* (Neural CoT) and *System 2* (Symbolic Solver) when they disagree ($r_{sym} \neq r_{neur}$). The **Socratic Arbiter** uses Neural Confidence ($C_{neur}$) and Program Entropy (NES) to decide which reasoning path is more reliable. Below are two examples showing different arbitration outcomes.

### A.2.1. SCENARIO A: THE TRIUMPH OF RIGOR (FAVORING SYSTEM 2)

In tasks requiring strict multi-hop deduction, neural models often hallucinate due to attention drift, resulting in low confidence. Here, the Arbiter trusts the Symbolic Solver because the logic program has low entropy (indicating a clear translation).

### A.2.2. SCENARIO B: THE SAFETY OF INTUITION (FAVORING SYSTEM 1)

In cases involving linguistic ambiguity or implicit common sense (which are hard to formalize), the Symbolic Solver may fail to prove the statement (returning "Uncertain") or rely on a shaky translation (High Entropy). If the Neural model is

---

**Case 2.1: Arbiter Favors Symbolic Solver**

**Input:** "All A are B. All B are C. No C are D. Is A a D?"
**System 1 (Neural):** "Yes." ($C_{neur} = 0.45$ – Low Confidence)
*Reasoning:* The model gets confused by the chain of negations.
**System 2 (Symbolic):** "False" (Strict Derivation: $A \rightarrow B \rightarrow C \rightarrow \neg D \therefore A \rightarrow \neg D$).
**Logic Program Entropy:** Low (Translation is straightforward).
**Socratic Arbitration:**
"Discordance detected. The Neural Confidence is low (0.45), suggesting uncertainty in the trace. The Logic Program is syntactically valid with low entropy. **Decision: Trust System 2 ($r_{sym}$).**"
**Final Output:** False.

---

highly confident, the Arbiter treats the logic program as "overly restrictive" and falls back to neural intuition.

---

**Case 2.2: Arbiter Favors Neural Intuition**

**Input:** "The sky is blue. Things that are blue are colored. Is the sky colored?"
**System 1 (Neural):** "Yes." ($C_{neur} = 0.98$ – Very High Confidence)
**System 2 (Symbolic):** "Uncertain."
*Reasoning:* The solver failed because the translation missing a specific bridge between "Things" and "Sky" or used an Open World Assumption.
**Logic Program Entropy:** High (Spikes on predicate mapping).
**Socratic Arbitration:**
"Discordance detected. The Symbolic Solver returned 'Uncertain', likely due to missing axioms. However, Neural Confidence is extremely high (0.98) and the logic relies on common sense. The logic program contains high-entropy mappings. **Decision: Trust System 1 ($r_{neur}$).**"
**Final Output:** Yes.

---

## B. Validity of Neural Confidence Estimation

Figure 4a to Figure 4e illustrate the distribution of the proposed neural confidence scores ($C_{neur}$) for both correct and incorrect predictions across all five datasets. We observe a clear separation: correct predictions tend to cluster towards higher confidence scores, while incorrect predictions are distributed towards the lower end. This empirical evidence validates that our neural confidence estimation effectively serves as a reliability indicator for the model's reasoning process.

## C. Experimental Setup Details

### C.1. Datasets

We evaluate our framework on four standard logical reasoning benchmarks, comprehensively covering a broad spectrum of reasoning types:

- ProntoQA (Saparov & He, 2022): A synthetic dataset focusing on multi-hop deduction with purely fictional predicates devoid of prior knowledge.

- ProofWriter (Tafjord et al., 2021): Requires generating explicit natural language proofs over N-hop rules, testing strict deductive capabilities.

- FOLIO (Han et al., 2024): A challenging dataset with expert-written First-Order Logic problems containing complex linguistic nuances and real-world knowledge.

- LogicalDeduction (Srivastava et al., 2023): A task from BigBench comprising Constraint Satisfaction Problems (CSP) that require deducing the order of objects from a minimal set of conditions.

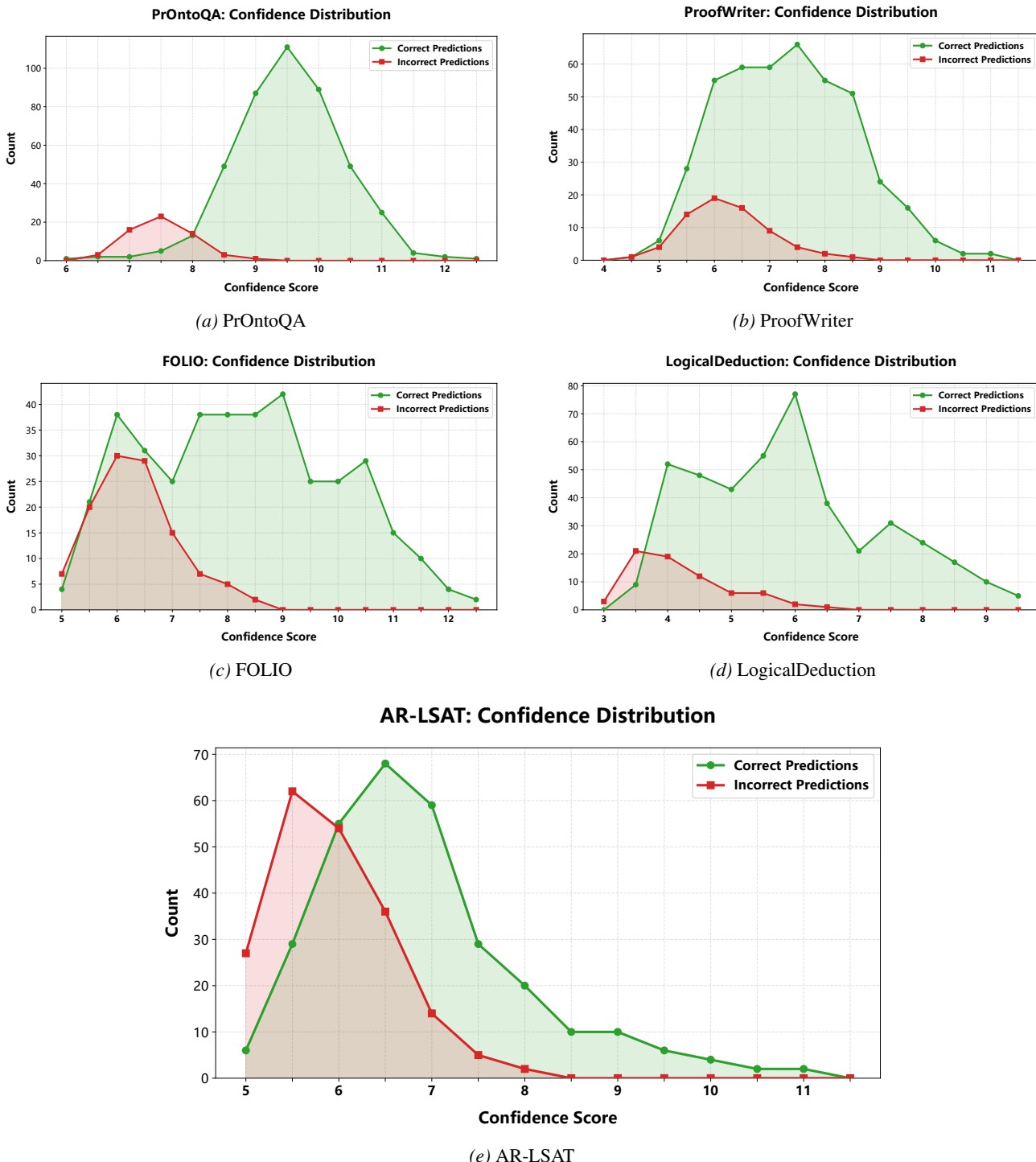

*Figure 4.* **Confidence distributions across five benchmarks.** Comparison between correct (blue) and incorrect (red) predictions across PrOntoQA, ProofWriter, FOLIO, LogicalDeduction, and AR-LSAT.

- AR-LSAT (Zhong et al., 2022): Analytical Reasoning questions from the Law School Admission Test, representing highly complex real-world constraint satisfaction problems (CSP) in out-of-distribution scenarios.

## C.2. Baselines

To rigorously evaluate the effectiveness of LogicSAGE, we compare it against a comprehensive set of baselines, categorized into two paradigms based on their reasoning methodology:

1. **Pure Neural Prompting Baselines**: These methods rely primarily on the internal reasoning capabilities of Large Language Models via advanced prompting strategies.

   - **GPT-4 (Standard)** and **GPT-4-CoT**: We report the performance of GPT-4 using both direct answering and Chain-of-Thought (CoT) prompting (Wei et al., 2022) to represent the upper bound of pure neural reasoning.
   - **SymbCOT** (Xu et al., 2024b): A strong baseline that augments the standard Chain-of-Thought with symbolic execution traces to guide the generation process, representing the state-of-the-art in prompt-based logical enhancement.

2. **Neuro-Symbolic Frameworks**: These methods, like ours, explicitly integrate LLMs with external solvers but differ in their architecture and refinement strategies.

   - **Logic-LM** (Pan et al., 2023): The foundational baseline that employs a standard generate-then-verify loop based on error feedback using GPT-4.
   - **Logic-LM++** (Kirtania et al., 2024): An enhanced version of Logic-LM that incorporates multi-step refinement and paired candidate comparisons to improve translation accuracy.
   - **LTRAG** (Hu et al., 2025): A Retrieval-Augmented Generation (RAG) framework that retrieves similar logical templates to serve as demonstrations to assist the solver, particularly effective for tasks requiring external knowledge (e.g., AR-LSAT).
   - **VERUS-LM** (Callewaert et al., 2025): A recent state-of-the-art modular neuro-symbolic solver that uses a specialized ensemble approach decomposing problems into sub-tasks for high-precision deduction.

## C.3. Implementation Details

To demonstrate the efficiency and democratized potential of our framework, we restrict the neural backbone to Qwen-3-8B [1], a parameter-efficient open-source LLM. Specifically, the Neural Semantic Parser, the Socratic Error Correction Agent, and the Neural Reasoning Chain (*System 1*) are all instantiated using the same Qwen-3-8B model, fine-tuned via LoRA on the training splits of the respective datasets. This unified setup ensures that our performance gains stem from architectural innovation rather than raw model scale. For symbolic execution, we align specific deterministic solvers with the logical topology of each benchmark: deductive reasoning tasks (PrOntoQA and ProofWriter) utilize the Pyke logic programming engine; First-Order Logic problems (FOLIO) are offloaded to the Prover9 automated theorem prover; Constraint satisfaction tasks (LogicalDeduction) employ the python-constraint library; and complex analytical reasoning tasks (AR-LSAT) are solved using the Z3 SMT solver. The Socratic Agent operates with a maximum refinement depth of $T_{max} = 3$. All experiments were conducted on a server with a single NVIDIA H100 (80GB) GPU.

## C.4. Socratic-Augmented SFT Data Construction

Standard distillation methods often discard failed generation attempts, resulting in a bias towards "easy" samples and a loss of valuable data on complex queries. To overcome this, we implemented a *Socratic-Augmented Distillation Pipeline*, enabling the teacher model (Qwen-3-Max[2]) to self-correct and synthesize high-fidelity logic programs even for hard-to-solve problems. The pipeline consists of three stages:

**1. Initial Teacher Generation.** For each training instance $(C, Q, y)$, we prompted the teacher model (Qwen-3-Max) to generate a candidate logic program $\Phi_0$ based on the domain schemas.

---

[1] https://huggingface.co/Qwen/Qwen3-8B
[2] https://chat.qwen.ai/

**2. Socratic Remediation Loop.** We executed $\Phi_0$ using the symbolic solver. If the program failed (due to syntax errors or label mismatch), we did not discard it. Instead, we triggered the *Dual-Mode Socratic Agent* (powered by the same `Qwen-3-Max` teacher):

- **Reactive Repair:** If the solver returned a `SyntaxError`, the teacher model was prompted with the error trace to generate a fixed version $\Phi_1$.

- **Proactive Refinement:** If the solver returned a result inconsistent with the ground truth $y$ (Label Mismatch), the teacher model was prompted to re-examine the logic for semantic misalignments.

This loop continued for a maximum of $T = 3$ turns or until the program was verified.

**3. Strict Verification & Retention.** We retained the final logic program $\Phi_{final}$ for the SFT dataset if and only if it passed the **Solver Verification Protocol**: (i) valid syntax, (ii) successful execution, and (iii) consistent label with ground truth.

**Outcome.** This strategy allowed us to "salvage" a significant subset of complex samples that were initially incorrectly generated, enriching the dataset with difficult reasoning paths. The final dataset consists of pairs $(X, \Phi_{verified})$, allowing the student model (`Qwen-3-8B`) to learn from the best-effort corrected reasoning traces.

## D. Entropy-Based Probing: Diagnostic Evaluation and Threshold Robustness

**Detection Performance.** Table 5 reports the precision and recall of the entropy spike trigger for detecting silent semantic failures. We explicitly separate program-level and error-level metrics: *Program Trigger Coverage* measures the fraction of faulty programs containing at least one spike, while *Precision/Recall* operate at the individual error instance level.

*Table 5.* Diagnostic evaluation of entropy-based proactive probing.

| Dataset | Prog. Trigger Coverage (%) | Prog. Non-Trigger Rate (%) | Precision (%) | Recall (%) | Error Miss (=1−Recall, %) |
|---|---|---|---|---|---|
| PrOntoQA | N/A (no silent failures observed) | | | | |
| ProofWriter | 67.2 | 32.8 | 80.4 | 55.6 | 44.4 |
| FOLIO | 71.4 | 28.6 | 77.1 | 58.3 | 41.7 |
| LogicalDed. | 69.1 | 30.9 | 82.3 | 59.4 | 40.6 |
| AR-LSAT | 83.6 | 16.4 | 75.8 | 65.7 | 34.3 |

The trigger achieves consistently high precision (75.8–82.3%), indicating that most flagged spikes correspond to genuine semantic errors. The error-level miss rate (34–44%) reflects confident-but-wrong translations where the model produces low entropy despite incorrect logic. Proactive probing is therefore best understood as a high-precision selective trigger that complements, rather than replaces, reactive correction.

**Threshold Robustness.** Our threshold $\alpha = 3$ is grounded in the $3\sigma$ rule rather than dataset-specific tuning. Table 6 reports a sensitivity sweep on AR-LSAT.

*Table 6.* Threshold sensitivity on AR-LSAT.

| $\alpha$ | **Trigger Rate** | **Precision** | **Recall** | **Accuracy (%)** |
|---|---|---|---|---|
| 2.0 | 98.5% | 38.6% | 89.1% | 62.5 |
| 2.5 | 92.8% | 62.4% | 78.9% | 74.3 |
| **3.0** | **83.6%** | **75.8%** | **65.7%** | **77.0** |
| 3.5 | 54.3% | 84.1% | 39.5% | 68.8 |
| 4.0 | 25.1% | 93.4% | 18.2% | 61.2 |

Performance remains stable across $\alpha \in [2.5, 3.5]$, with severe degradation at the extremes: over-sensitive triggering ($\alpha \leq 2.0$) introduces false alarms, while over-conservative thresholds ($\alpha \geq 4.0$) miss the majority of errors.

## E. Detailed Prompt Templates

In this section, we present the comprehensive suite of prompt templates used in **LogicSAGE**. Following our Dual-Process narrative, the interaction is structured as a dialectic between a **Socratic Teacher Agent** (responsible for pedagogical

scaffolding, diagnostic inquiry, and proactive probing) and a **Socratic Student Agent** (responsible for formal logic generation and reflexive repair).

### E.1. Phase 1: Domain-Specific Neural Semantic Parsing ($\mathcal{T}_{schema}$)

The parser utilizes specialized templates for different logical topologies. These prompts enforce strict "Syntax Validation Constraints" to minimize initial execution failures.

#### E.1.1. TEMPLATE FOR ANALYTICAL REASONING (AR-LSAT / Z3 SOLVER)

---

**F.1.1 Semantic Parsing: AR-LSAT (Z3 Domain)**

**[System - Socratic Teacher Agent]**
You are a meticulous symbolic reasoning expert. Your goal is to guide the student to formulate the following problem into a logic program consisting of `Declarations`, `Constraints`, and `Options`.
**CORE REQUIREMENTS & CONSTRAINTS:**

- **Declarations**: Declare ALL sorts and functions BEFORE using them. Use `EnumSort`.

- **Syntax Validation**:
    - **Logical Operators**: Use `And, Or, Not, Implies, ForAll, Exists, Count, Distinct`. Never use symbols like `&, |, !, ∧`.
    - **Arithmetic**: Use `+, -, *, /` ONLY for `int` types. For adjacency, use `Abs(x - y) == 1`.
    - **Equality**: Use `==` for logical comparison, not `=`.

- **Formatting**: Every constraint must use the `FORMULA ::: justification` format.

- **Options**: Consistently use `is_sat(FORMULA)` or `is_valid(FORMULA)`. For exceptions, use `is_exception(is_sat(FORMULA))`.

**[User - Input]** Problem: {Context} Question: {Question} Choices: {Options}
**[Socratic Student Response]**
# Declarations
...
# Constraints
...
# Options
...

---

#### E.1.2. TEMPLATE FOR FIRST-ORDER LOGIC (FOLIO)

---

**F.1.2 Semantic Parsing: FOLIO (Prover9 Domain)**

**[System - Socratic Teacher Agent]**
Your task is to parse the problem into First-Order Logic (FOL) formulas.
**ENTITY & PREDICATE NAMING CONSTRAINTS:**

- ALL individuals MUST use lowercase/camelCase (e.g., `rina`, `james`, `zeldaCreator`).

- Use descriptive PascalCase predicates (e.g., `Dependent(x)`, `Author(x, y)`).

- Use standard grammar: Conjunction (∧), Disjunction (∨), XOR (⊕), Negation (¬), Implication (→), Biconditional (↔).

**[Output Format]**
`Predicates: — Premises: — Conclusion:`

---

## E.2. Phase 2: Mode 1 - Reactive Socratic Dialectic (Error Correction)

Triggered when the symbolic solver returns a traceback ($\epsilon$), this prompt models the "Student-Teacher" debugging session.

---

**F.2.1 Mode 1: Reactive Socratic Debugging Loop**

**[Socratic Teacher Agent]**
I have evaluated your logic program, but the Symbolic Solver encountered a terminal `Error_Type`: `Error_Message`.
**Pedagogical Feedback for Repair:** 1. **Localization**: The failure occurred near the token 'Error_Token'. 2. **Naming Check**: Did you declare this variable? Ensure names like 'Nail' and 'Nails' are not mixed. 3. **Function Arity**: Check the function declaration `Function_Name`. Does the number of arguments in `# Constraints` match the definition? 4. **Syntax Validation**: Are you using `is_sat` instead of `IsSat`? Ensure all variables in `Count` are bracketed: `Count([x:sort], ...)`.
**Context History**: Failed_Attempts_List
Please provide the evolved logic program that corrects these syntactic misalignments.
**[Socratic Student Agent]**
I acknowledge the syntactic fragility. I see that I failed to declare the sort for 'Token' and used an incorrect operator. I will fix the `# Declarations` and re-generate the program.

---

## E.3. Phase 3: Mode 2 - Proactive Socratic Probing (Entropy-Aware)

Triggered by low-confidence tokens (high entropy spikes), the Teacher questions potential "Silent Failures".

---

**F.3.1 Mode 2: Proactive Uncertainty Probing**

**[Socratic Teacher Agent]**
Your logic program is syntactically valid and executed successfully. However, I have detected **Normalized Entropy Spikes** (Confidence Threshold) on specific logical constructs.
**Identified Uncertainty Spikes:**

- **Predicate**: 'Token_1' (Confidence Score: Prob_1)

- **Connective**: 'Token_2' (Confidence Score: Prob_2)

**Socratic Inquiry regarding Semantic Soundness:** Carefully re-examine the premise segment: "Natural_Language_Segment". - You translated "Neither...nor" as `Current_Logic`. In this context, does "Neither" imply a conjunction of negations? - Is there a potential singular/plural mismatch or entity ambiguity here?
Provide a **Predicate Analysis** and, if a semantic misalignment is confirmed, provide a **Corrected Logic Program**.
**[Socratic Student Agent]**
**Analysis**: Upon re-reading the premise, I realize that 'Neither the wolf nor the fox' was incorrectly mapped to a simple conjunction. This is a semantic hallucination. **Refined Logic**: ...

---

## E.4. Phase 4: System 1 - Neural Reasoning with Formal Grounding

This prompt forces the neural intuitive reasoner to follow the formal logic program ($\Phi$) as an axiomatic skeleton.

---

**F.4.1 Neural Reasoning with Formal Skeleton**

**[System Instruction]**
You are a meticulous reasoning assistant. Rely ONLY on the provided problem context and its **Formal Symbolic Skeleton** ($\Phi$) to determine the answer.
**[Input Data]**
Context ($C$): Context
Formal Skeleton ($\Phi$):
`Logic_Program`
**Reasoning Protocol:** 1. Your thinking process must explicitly reference axioms from $\Phi$ (e.g., "According to `Constraint_1`, we know..."). 2. Do not deviate from the logical boundaries established in the skeleton. 3. Perform a step-by-step deduction (CoT).
**[Neural Response]**
Thinking Process: ...
Answer: ...

---

## E.5. Phase 5: Adaptive Fusion and Arbitration

This template resolves conflicts when the Symbolic result ($r_{sym}$) and Neural result ($r_{neur}$) diverge.

---

**F.5.1 Socratic Arbitration (Conflict Resolution)**

**[System - Arbiter Persona]**
We have a discordance between our dual thinking processes: - **System 2 (Symbolic Solver)** derived: 'r_sym'. - **System 1 (Neural CoT)** predicted: 'r_neur' with Confidence $C_{neur}$: Value.
**Arbitration Task**: Review the Logic Program $\Phi$. Did the solver arrive at 'r_sym' because the logic was **overly restrictive** or misinterpreted the core semantic intent of the premise? - If $C_{neur}$ is high and the Symbolic branch has low-confidence predicates (Mode 2 Spikes), the Neural intuition may be more robust. - If $C_{neur}$ is low, favor the rigorous symbolic proof.
Final Decisive Answer: ...

---

