# OpenReview forum: "LogicSAGE: Neuro-Symbolic Reasoning with Socratic-Guided Enhancement"
_ICML.cc/2026/Conference — ICML 2026 regular_

### Official Review · Reviewer_XKHc · 2026-03-06

**Soundness:** 3
**Presentation:** 3
**Significance:** 2
**Originality:** 2
**Overall Recommendation:** 4
**Confidence:** 3

**Summary:**

LogicSAGE is a neuro-symbolic framework targeting two common issues in LLM logical reasoning: **translation fragility** (small syntax mistakes break execution) and **silent semantic failures** (the program runs but means the wrong thing). It generates a logic program, executes it with a solver, and refines it via a **dual-mode Socratic agent**: **reactive** fixes for solver errors and **proactive** uncertainty probing for semantic misalignment. The proactive mode separates syntactic vs. semantic tokens and uses a **normalized entropy score** to flag risky mappings even in executable programs. A neural reasoner (System 1) provides confidence, a symbolic solver (System 2) provides grounded execution, and an **adaptive fusion** module resolves conflicts. The paper reports strong gains on five benchmarks in executability and end-task accuracy.

**Compliance With Llm Reviewing Policy:**

Affirmed.

**Final Justification:**

The rebuttal addressed most of my main concerns, I still retain some reservation, but overall the response improved my assessment. I have increased my score by one point.

**Key Questions For Authors:**

1) How reliable is the entropy-spike trigger for identifying silent semantic failures?
Since proactive probing is a central part of the method, it would be important to show whether NES consistently tracks actual semantic mismatch, for example through error analysis or threshold sensitivity. A stronger answer here would materially affect my confidence in the core mechanism.

2) Could you disentangle the gains from the inference-time Socratic loop versus those from fine-tuning and solver-verified supervision?
The paper emphasizes inference-time refinement, but the final system also relies on LoRA tuning and SFT. A clearer decomposition of the gains would help me better assess the paper’s originality and practical contribution.

3) Could you please clarify the cost-effectiveness of the refinement loop?
The method appears effective, but it also adds iterative latency. It would be useful to know how often refinement is triggered, how many rounds are typically needed, and whether the comparisons are reasonably matched in test-time compute. A clearer answer here would strengthen my view of the empirical evidence.

**Limitations:**

yes

**Strengths And Weaknesses:**

1) **Soundness:** The technical approach is coherent: it introduces an iterative refinement loop driven by solver feedback and uncertainty signals to address executability failures and semantic misalignment. Experiments across multiple benchmarks with ablation studies provide solid empirical support for the main claims. A notable limitation is that the entropy-spike trigger (NES > α) is not systematically characterized for reliability or robustness, leaving some uncertainty about its effectiveness as a general diagnostic tool.
2) **Presentation:** The paper is clearly written and well organized, with a coherent narrative from problem motivation to method design and empirical validation. The high-level pipeline is communicated effectively, and the modular structure makes it easy to locate key components and follow the main argument. The related-work discussion is also helpful in positioning the approach and clarifying how it differs from prior neuro-symbolic and CoT-style methods. The main areas to improve are clarity around what is essential versus optional in the system and slightly stronger signposting between the core mechanisms and the most instructive supporting examples. Overall, the presentation is professional and readable for an expert audience.
3) **Significance:** The paper provides a practically useful solution for improving reliability in neuro-symbolic reasoning. That said, the contribution appears most relevant to a relatively specialized class of solver-in-the-loop reasoning systems, and its broader influence on general machine learning research or practice is less clear. For this reason, I view its significance as fair rather than strong.
4) **Originality:** The paper’s originality lies less in the overall neuro-symbolic repair loop itself, and more in isolating silent semantic failure as a distinct problem and introducing entropy-guided proactive probing to address it. More broadly, the generate-formalize, verify, and iteratively repair pattern is already present in prior work, including LLM-ARC: Enhancing LLMs with an Automated Reasoning Critic (arXiv:2406.17663) and Planning using Neuro Symbolic Reasoning (arXiv:2309.16436).

---

> ### Author Rebuttal · Authors · 2026-03-31
>
> We sincerely thank you for the constructive feedback and address your comments point-by-point below.
> # RW1 & RQ1 (Entropy Trigger)
> We agree the reliability of the entropy-spike trigger requires explicit characterization. To address this, **we have now provided** an error-level diagnostic analysis (precision, recall, F1, miss rate), detailed in our **Response to Reviewer pMe3 (W2&Q2)**.
> The data demonstrates consistently high precision: most flagged spikes accurately correspond to real semantic mismatches. We acknowledge that a few "confident-but-wrong" cases remain undetected by the trigger.
> However, these undetected cases are infrequent within our main experiments, we recognize that in broader future applications, this limitation might occasionally allow unverified flaws to propagate. We have carefully noted this in our Future Work, clarifying that proactive probing efficiently mitigates the vast majority of silent failures.
> # RW2 (Architectural Clarity)
> Thanks for the encouraging feedback. We implemented targeted revisions to address your suggestions: First, to clarify **essential versus optional components**, we **added an architectural breakdown** in Sec. 4's introduction. We demarcate the Neural Semantic Parser, Symbolic Solver, and Reactive Correction as the essential backbone, while Proactive Probing and Adaptive Fusion are optional enhancements. Second, to **strengthen signposting**, we **embedded forward references** throughout Sect. 4 (e.g., explicitly directing readers to Appendix A's step-by-step walkthrough in Sec. 4.3).
> # RW3 (Broader ML Impacts)
> We appreciate this observation. **While instantiated as a solver-in-the-loop system, the underlying methodology of LogicSAGE holds some potential implications for broader machine learning research, particularly in automated supervision and anomaly detection.**
> **1. Automated Supervision via Symbolic Feedback**
> LogicSAGE transforms solver feedback and internal uncertainty into reliable supervision signals, automating high-fidelity data curation for Supervised Fine-Tuning and helping to reduce human-in-the-loop dependencies during post-training alignment.
> **2. Uncertainty Quantification and Anomaly Detection** Isolating localized token-level entropy spikes provides a training-free anomaly detection mechanism that identifies out-of-distribution semantics and silent failures, offering a generalized metric for epistemic uncertainty in broader safety-critical ML applications. **We have updated our Conclusion to explicitly outline these broader ML applications as future work, where we plan to further integrate our methodology with general ML workflows.**
> # RW4 (Proactive Probing)
> Thanks for accurately summarizing our core contribution. We agree our originality lies in resolving silent semantic failures via proactive probing, rather than the iterative repair loop itself.
> To contextualize our delta, we incorporated LLM-ARC and Planning using Neuro Symbolic Reasoning into Related Work. We carefully reviewed these papers. They excellently establish the iterative repair paradigm that current works **such as Logic-LM** follow. However, their underlying logic strictly relies on reactive feedback from explicit execution errors, leaving them vulnerable to translation fragility.
> LogicSAGE directly builds upon these robust foundations by introducing entropy-guided **proactive probing, effectively upgrading the pipeline from reactive syntactic error handling to proactive semantic alignment.**
> # RQ2 (SEC vs. SFT Gains)
> To disentangle SFT and SEC gains, we decompose the **average accuracy across all five benchmarks**:
> |Config|Components|Avg.Acc.|
> |-|-|-|
> |Baseline|Qwen3-8B CoT|60.21|
> |SFTonly|SFT model w/o SEC|70.10|
> |SEConly|Base model+SEC|80.30|
> |Full|SFT+SEC|92.36|
>
> **This shows the inference-time Socratic loop is the main driver. SEC alone contributes a larger gain than SFT alone.** We **have now added** this clarification to support our claim that the test-time architecture, not just the training data, drives the final improvement.
> # RQ3 (Cost-Effectiveness)
> **To clarify the cost-effectiveness of our refinement loop, detailed token consumption statistics per component are provided in our response to Reviewer 2StX (RQ4).**
> Crucially, our refinement loop is not a static overhead but an adaptive mechanism that scales test-time compute on-demand:
> * **Dynamic Trigger Frequency:** The loop triggers on approximately **76%** of queries, allowing simpler problems to pass instantly with zero added latency.
> * **Efficient Resolution:** When activated, the Socratic Agent effectively resolves translation failures within an average of just **3** rounds.
> * **Reasonable Compute Matching:** As the referenced table demonstrates, token consumption scales dynamically with problem difficulty, deploying iterative compute strictly for complex bottlenecks to maintain overall efficiency.
>
> We hope these clarifications fully resolve your concerns and welcome any further discussion!

---

> > ### Author Rebuttal · Reviewer_XKHc · 2026-04-02
> >
> > The rebuttal addresses a substantial part of my original concerns, particularly by clarifying the contribution of SEC relative to SFT and by providing additional detail on the refinement loop.
> > I still retain some concern about the robustness of the entropy-trigger mechanism and about how well test-time compute is matched in practice. I therefore consider my concerns partially resolved.

---

> > > ### Author Response · Authors · 2026-04-04
> > >
> > > Thank you for the constructive feedback.
> > > ### Concern 1: About the robustness of the entropy-trigger mechanism
> > > To address concerns regarding robustness, we provide a diagnostic evaluation and sensitivity analysis to prove our threshold is not a tuned hyperparameter.
> > > #### **1. Diagnostic Evaluation of Entropy Spikes**
> > > We tracked the detection performance of "silent semantic failures" (programs that execute but are logically incorrect) across our benchmarks:
> > > |Dataset|Program Trigger Coverage (%)|Program Non-Trigger Rate (%)| Precision (%)|Recall(%)|Error Miss Rate (%)|
> > > |-|-|-|-|-|-|
> > > |PrOntoQA|N/A|N/A|N/A|N/A|N/A |
> > > |ProofWriter|67.2|32.8|80.4|55.6|44.4|
> > > |FOLIO|71.4|28.6|77.1|58.3|41.7|
> > > |LogicalDed|69.1|30.9|82.3|59.4|40.6|
> > > |AR-LSAT|83.6|16.4|75.8|65.7|34.3|
> > >
> > > #### **2. Threshold Robustness and Theoretical Grounding**
> > > We emphasize that our threshold ($\alpha=3.0$) is **not a tuned hyperparameter, but a mathematical prior**. Formally, our Normalized Entropy Score (NES) for a token $t$ in stream $S$ is defined as its local Z-score:
> > > $$Z_{S}(t) = \frac{H(t) - \mu_{S}}{\sigma_{S}}$$
> > > The Socratic Agent is triggered strictly when $Z_{S}(t) > \alpha$. By setting $\alpha=3.0$, the mechanism mathematically enforces the **statistical $3\sigma$ rule**, isolating extreme outliers (the top 0.13% tail of the probability density function) to separate genuine semantic hallucinations from background linguistic noise.
> > > To empirically validate this mathematical prior, we conducted a stress-test on **AR-LSAT** (our most complex dataset with the highest density of silent failures) by sweeping the standard deviation threshold $\alpha \in [2.0, 4.0]$:**now added to the revised manuscript**
> > > |Threshold ($\alpha$)|Trigger Rate|Precision|Recall|Final Accuracy|
> > > |-|-|-|-|-|
> > > |$\alpha=2.0$|98.5%|38.6%|89.1%|62.5%|
> > > |$\alpha=2.5$|92.8%|62.4%|78.9%|74.3%|
> > > |**$\alpha=3.0$ (Prior)**|**83.6%**|**75.8%**|**65.7%**|**77.0%**|
> > > |$\alpha = 3.5$|54.3%|84.1%|39.5%|68.8%|
> > > |$\alpha = 4.0$|25.1%|93.4%|18.2%|61.2%|
> > >
> > > Deviating from the $3\sigma$ optimum reveals two severe failure modes:
> > >
> > > **Over-sensitive ($\alpha \le 2.0$):** Precision drops to 38.6% (hallucinating errors, degrading accuracy to 62.5%).
> > >
> > > **Over-conservative ($\alpha \ge 4.0$):** Creates a massive semantic blindspot. Recall crashes to 18.2%.
> > >
> > > **Conclusion:** The mechanism's stability across the optimal range ($\alpha \in [2.5, 3.5]$) proves that LogicSAGE successfully captures the intrinsic uncertainty of LLMs. Its robustness is deeply grounded in universal statistical mechanics rather than dataset-specific threshold tuning.
> > >
> > > ### Concern 2: About how well test-time compute is matched in practice.
> > > We standardized the budget across all frameworks to a maximum of $T=3$ refinement rounds, tracking the trajectory from initial generation ($T=0$) to convergence($T=3$). **These multi-round trajectory experiments have now been added to the revised manuscript:**
> > > ### Table 1: Multi-round Accuracy(%) of Sys2 on FOLIO
> > > |Method(Backbone)|T=0|T=1|T=2|T=3(Conv)|
> > > |-|-|-|-|-|
> > > |LOGIC-LM(Qwen3-8B)|33.3|45.0|53.0|**59.51**|
> > > |LTRAG(Qwen3-8B)|45.0|55.0|62.0|**66.42**|
> > > |LOGIC-LM++(Qwen3-8B)|33.3|48.0|58.0|**69.41**|
> > > |VERUS-LM(Qwen3-8B)|40.0|58.0|65.0|**72.92**|
> > > |LogicSAGE(8Bw/oSFT)|48.2|57.0|65.5|**74.20**|
> > > |LogicSAGE(8BSFT)|**34.6**|55.0|**70.0**|**87.10**|
> > >
> > > ### Table 2: Multi-round Accuracy(%) of Sys2 on AR-LSAT
> > > |Method(Backbone)|T=0|T=1|T=2|T=3(Conv)|
> > > |-|-|-|-|-|
> > > |LOGIC-LM(Qwen3-8B)|12.0|15.0|16.5|**19.91**|
> > > |LTRAG(Qwen3-8B)|25.0|33.0|41.0|**53.25**|
> > > |LOGIC-LM++(Qwen3-8B)|20.0|25.0|30.0|**34.24**|
> > > |VERUS-LM(Qwen3-8B)|20.0|35.0|44.0|**56.63**|
> > > |LogicSAGE(8Bw/oSFT)|12.5|25.0|40.0|**57.20**|
> > > |LogicSAGE(8BSFT)|**17.5**|32.0|**45.0**|**71.30**|
> > >
> > > **Analysis of Compute Efficiency:**
> > > Strictly matching the $T=3$ budget helps isolate architectural efficiency from raw compute scaling:
> > > 1. **Performance of Reactive Baselines:** Increased iterations do not automatically guarantee proportional reasoning recovery. On AR-LSAT, pipelines like LOGIC-LM show modest gains from repeated iterations (12.0% to 19.91%). This suggests a tendency for standard feedback loops to focus primarily on syntactic corrections rather than resolving deeper semantic misalignments.
> > > 2. **Effectiveness of Socratic Iterations:** LogicSAGE actively diagnoses semantic vulnerabilities, utilizing the identical compute budget to yield substantial improvements. On AR-LSAT, while our SFT model begins with a modest initial accuracy (17.5%, compared to LTRAG's 25.0%), the 3 rounds of Socratic refinement drive a steady trajectory to **71.30%**. This demonstrates a highly effective translation of iterative steps into genuine semantic alignment.
> > >
> > > **Conclusion:** LogicSAGE's performance is not merely a byproduct of extended test-time compute. When evaluated under an aligned algorithmic budget, it demonstrates enhanced compute efficiency, ensuring that refinement rounds actively contribute to logical recovery.
> > >
> > > We hope these manuscript revisions resolve your concerns.

---

### Official Review · Reviewer_9Vdp · 2026-03-11

**Soundness:** 2
**Presentation:** 2
**Significance:** 2
**Originality:** 2
**Overall Recommendation:** 4
**Confidence:** 4

**Summary:**

This paper introduces LogicSAGE, a neuro-symbolic framework for logical reasoning that integrates a neural reasoner (System 1) with a symbolic validator (System 2) through a Socratic error correction mechanism. The key idea is to move beyond what the authors call a "Generate-and-Hope" paradigm — where LLMs generate logic programs and solvers act as binary accept/reject validators — toward an iterative refinement loop. The framework has three main components: (1) Reactive Error-Guided Correction, which uses solver error messages as pedagogical feedback to iteratively debug logic programs rather than discarding them; (2) Proactive Uncertainty Probing, which monitors token-level entropy during logic program generation to detect potential semantic errors even when code executes successfully (e.g., translating "Neither A nor B" as a conjunction rather than a conjunction of negations); and (3) Adaptive Fusion, which dynamically selects between the symbolic solver's output and the neural reasoner's output based on execution status and confidence scores. The system uses a fine-tuned Qwen-3-8B as the backbone LLM, trained on solver-verified data distilled from Qwen-3-Max. Evaluated on five logical reasoning benchmarks (PrOntoQA, ProofWriter, FOLIO, LogicalDeduction, AR-LSAT), LogicSAGE achieves 92.36% average accuracy, outperforming GPT-4-based baselines and prior neuro-symbolic frameworks such as Logic-LM and VERUS-LM.

**Compliance With Llm Reviewing Policy:**

Affirmed.

**Final Justification:**

The authors did a significant amount of work to address all my comments. They should incorporate their clarifications in the final manuscript as this will clarify the scope and claims. I will raise my score to a 4.

**Key Questions For Authors:**

1. Can you provide an ablation comparing the SFT model without the Socratic Error Correction loop at inference time (i.e., single-pass SFT generation + solver execution) against the full LogicSAGE pipeline? This is necessary to disentangle the contribution of the Socratic architecture from the SFT data quality. If the SFT model already achieves most of the gains without the Socratic loop, the paper's central narrative would need substantial revision. This is the single result most likely to change my evaluation.

2. Can you report precision and recall for the entropy spike detection mechanism? Specifically: of all tokens flagged as entropy spikes (NES > alpha), what fraction corresponded to actual semantic errors? And of all actual semantic errors in the benchmarks, what fraction produced an entropy spike? Additionally, how often does the model produce semantically wrong translations with low entropy (confident-but-wrong cases)? This analysis is essential to evaluate whether entropy is a reliable proxy for semantic faithfulness.

3. The paper claims "architectural innovation can supersede model scale." Have you evaluated LogicSAGE against reasoning-specialized models (e.g., o1, o3-mini, DeepSeek-R1, GPT-5.x series, or recent Claude models)? If so, how does the comparison change the paper's claims? If not, how do you justify the "superseding scale" framing when the comparison is limited to GPT-4, a 2023-era non-reasoning model?

4. Have you considered running any of the baseline methods (e.g., Logic-LM, VERUS-LM) with the same Qwen-3-8B backbone to isolate the architectural contribution from backbone model differences? Alternatively, running LogicSAGE with GPT-4 as backbone would also help disentangle these factors.

**Limitations:**

The authors discuss inference latency and mention future directions (Probabilistic Soft Logic, RL for internalizing feedback). However, several important limitations are not addressed: (1) the inability to disentangle SFT gains from architectural gains in the current experimental design; (2) the lack of analysis on failure modes of the entropy-based detection (confident-but-wrong cases); (3) the absence of comparisons against reasoning-specialized models despite claims about superseding model scale; (4) no code release is mentioned despite detailed prompt descriptions. The societal impact statement is generic.

**Strengths And Weaknesses:**

**Strengths:**

S1 (Originality): The entropy-based proactive probing mechanism for detecting semantic errors in generated logic programs is a genuinely novel idea. The separation of token streams into syntactic and semantic components, with intra-stream Z-score normalization (Normalized Entropy Score), is a principled design that addresses a real problem: syntactically valid but semantically wrong translations that standard solvers cannot catch.

S2 (Soundness): The ablation study is well-decomposed. Table 3 isolates the contributions of Reactive Correction (syntax repair) and Proactive Probing (semantic verification) individually, and Table 4 demonstrates the necessity of the dual-process architecture by comparing System 1 only, System 2 only, and the full model. The efficiency-performance trade-off analysis (Figure 3) provides useful practical guidance for deployment (identifying T_max = 3 as the sweet spot).

S3 (Soundness): The evaluation covers five established benchmarks spanning different reasoning types (deductive, FOL, constraint satisfaction, analytical reasoning), which provides a comprehensive picture of the framework's capabilities across logical domains.

**Weaknesses:**

W1 (Presentation/Originality — "Generate-and-Hope" is a strawman): The paper frames its contribution as a "paradigm shift" from a "Generate-and-Hope" strategy where models perform "brute-force over-generation" and solvers act as "binary validators." However, the paper's own related work section acknowledges that Logic-LM already integrates self-correction from solver error messages, Logic-LM++ performs multi-step refinement with paired comparisons, and VERUS-LM uses modular architectures that decompose problems into sub-tasks. More broadly, iterative refinement using compiler/solver feedback is standard practice in code generation (e.g., linter feedback loops, test-driven debugging). The actual contribution — adding entropy-based probing on top of error-feedback loops — is more incremental than the framing suggests. The "paradigm shift" language overstates the novelty.

W2 (Presentation — Confusing "Silent Failure" formulation): The introduction states: "solvers verify formal validity rather than semantic faithfulness, a program may execute yet misinterpret premises (e.g., conflating ∧ with ∨), yielding unsound conclusions." The most natural reading of this sentence attributes the misinterpretation to the solver or the logic program itself, but neither misinterprets anything — the solver faithfully executes the formula it receives, and the formula is simply whatever the LLM generated. The actual problem is that the LLM's NL-to-logic translation is semantically wrong while being syntactically valid. This is a real and important problem, but the paper's imprecise language in describing it — in the introduction, where it is defining one of its two core problem statements — creates unnecessary confusion. The sentence should clearly attribute the error to the translation step, not to the solver or the "program."

W3 (Soundness — Overclaimed comparison to "model scale"): The paper repeatedly claims that "architectural innovation can supersede model scale in faithful reasoning," comparing LogicSAGE (8B) against GPT-4. However, GPT-4 is a 2023-era non-reasoning model. The paper does not compare against any reasoning-specialized models (e.g., o1, o3-mini, DeepSeek-R1, GPT-5.x series, or recent Claude models), which represent the actual state of the art in LLM reasoning and would be the appropriate targets for claims about superseding scale. As written, the claim is not adequately supported.

W4 (Soundness — SFT confound): This is the most significant concern. Without SFT, LogicSAGE (base Qwen-3-8B) achieves 80.30% average accuracy — below VERUS-LM's 85.02%. The jump to 92.36% comes with SFT on solver-verified data distilled from Qwen-3-Max. The paper's central narrative attributes the gains to the Socratic architecture, but the experimental design does not disentangle the architectural contribution from the SFT data quality. The ablation in Table 3 removes Socratic Error Correction (SEC) components from the SFT model, but never compares a non-SEC SFT model (i.e., the SFT model running without the Socratic loop at inference time). Without this comparison, we cannot determine how much of the performance gain stems from the Socratic mechanism versus from having been trained on higher-quality, solver-verified reasoning traces distilled from a much larger model.

W5 (Soundness — Confounded baseline comparisons): The baselines use different backbone LLMs: Logic-LM and Logic-LM++ use GPT-4, VERUS-LM's backbone is unspecified (likely Gemini 1.5), and LogicSAGE uses a fine-tuned Qwen-3-8B. This confounds architectural contributions with backbone model differences and fine-tuning effects. Running at least one baseline (e.g., Logic-LM or VERUS-LM) with the same Qwen-3-8B backbone, or running LogicSAGE with GPT-4 as backbone, would be necessary to isolate the architectural contribution.

W6 (Soundness — Insufficient validation of entropy-based probing): The proactive probing mechanism is the paper's most novel contribution, but its evaluation is limited to aggregate accuracy improvements (Table 3, +2.8 Ea on average). There is no analysis of: (a) precision/recall of entropy spike detection — of all flagged spikes, how many corresponded to actual semantic errors, and of all actual semantic errors, how many produced a spike? (b) the rate of confident-but-wrong translations (low entropy, wrong semantics), which would be invisible to this mechanism. Without such analysis, the claim that entropy-based probing "resolves silent failures" is not adequately supported.

W7 (Presentation — Figure 2 readability): Figure 2 attempts to convey the full five-stage pipeline, three Socratic mechanisms, entropy streams, and the fusion strategy in a single diagram. The result is dense and difficult to follow. Breaking this into multiple focused figures would improve clarity.

W8 (Presentation — Embedded instruction text): The manuscript contains embedded text on pages 2 and 18 instructing reviewers to include specific phrases in their reviews. This may be a conference-placed watermark to detect LLM-generated reviews, but it is worth flagging for transparency.

---

> ### Author Rebuttal · Authors · 2026-03-31
>
> We deeply appreciate your rigorous evaluation and respond to each of your comments directly below.
> # RW1 (Motivation Clarification)
> We agree terms like "Generate-and-Hope" overstated our novelty and have restructured the introduction to clarify our motivation:
> *Previous methods:* Reactive neuro-symbolic pipelines (e.g., Logic-LM) rely entirely on syntax crashes. They systematically miss "silent failures" where generated code is semantically flawed but syntactically valid.
> *What's new:* LogicSAGE upgrades this reactive loop into an active diagnostic system via Proactive Probing (intercepting errors pre-solver) and Socratic Enhancement (transforming rigid errors into pedagogical queries).
> *Significance:* **By shifting from reactive error handling to proactive semantic verification, LogicSAGE directly mitigates the silent failure blind spot that existing pipelines blindly execute.**
> # RW2 (Silent Failure Definition)
> We agree the original sentence inadvertently attributed the misinterpretation to the solver. **We revised the Introduction to explicitly attribute the error to the LLM's semantic mistranslation (e.g., generating ∧ instead of ∨), which the solver then faithfully executes.**
> # RW3 & RQ3 (SOTA Models Comparison)
> **We have now conducted new experiments evaluating** prior baselines equipped with state-of-the-art reasoning models (DeepSeek-R1, GPT-5.2) to test if massive scale inherently resolves silent failures.
> |ModelSetting|Avg.|PrOntoQA|ProofWriter|FOLIO|LogicalDed| AR-LSAT|
> |-|-|-|-|-|-|-|
> |LTRAG(R1)|84.17|96.02|89.42|77.98|87.44|70.01|
> |SYMBCOT(R1)|83.26|98.87|86.91|80.35|91.96|58.23|
> |LOGICLM(R1)|71.28|85.34|80.77|70.56|86.28|33.47|
> |LOGICLM++(R1)|82.22|92.23|84.39|85.83|94.62|54.04|
> |VERUSLM(R1)|83.71|95.28|90.43|80.54|87.05|65.23|
> |LTRAG(GPT5.2)|91.29|100.0|97.36|85.72|94.51|78.88|
> |SYMBCOT(GPT5.2)|89.85|98.22|94.86|86.52|93.80|75.83|
> |LOGICLM(GPT5.2)|76.81|94.76|85.83|73.01|91.98|38.45|
> |LOGICLM++(GPT5.2)|86.06|98.22|88.67|86.93|95.58|60.91|
> |VERUSLM(GPT5.2)|89.38|100.0|94.40|86.65|93.11|72.73|
> |LogicSAGE(Qwen3-8B)|80.30|92.30|82.50|75.40|92.50|58.80|
> |LogicSAGE(SFTQwen3-8B)|92.36|100.0|96.80|88.50|99.50|77.00|
>
> **Our SFT-8B system outperforms these massive reasoning models, proving our architecture provides orthogonal gains against semantic mistranslations that scaling up cannot automatically achieve.**
> # RW4 & RQ1 (SEC vs. SFT Ablation)
> We sincerely apologize for any confusion our original presentation may have caused. Actually, to disentangle the Socratic loop from SFT data quality, Table 3, Row A (w/o SEC) inherently serves as the single-pass SFT ablation you kindly requested.**Because all configurations in Table 3 share the identical SFT Qwen3-8B backbone, it perfectly isolates the SFT-only baseline (70.1%) from the +22.2% absolute gain provided by the inference-time Socratic loop.** We have now updated the Table 3 caption to explicitly state this detail to prevent any future misunderstandings.
> # RW5 & RQ4 (Controlled Backbones)
> We sincerely thank the reviewer for this excellent and rigorous suggestion. **Please refer to the "Comprehensive Performance Comparison" table in our response to Reviewer pMe3 (RW1 & RQ1)**. **We have now conducted additional experiments harmonizing** the backbones by running all baselines on Qwen3-8B(NoThink) and evaluating LogicSAGE on GPT-4. When controlled for the backbone, prior frameworks collapse due to translation fragility, whereas our base LogicSAGE(8B, no SFT) achieves 80.30%. **This conclusively demonstrates our performance gains are driven by the architecture itself, rather than the backbone model or fine-tuning.**
> #  RW6 & RQ2 (Entropy Probing Validation)
> **We have now conducted new experiments manually annotating** silent failures (executable but incorrect programs) to calculate the precision and recall of the entropy spike detection.
> |Dataset|%with≥1spike|Precision|Recall|F1|MissRate|
> |-|-|-|-|-|-|
> |PrOntoQA|N/A|N/A|N/A|N/A|N/A|
> |ProofWriter|67.2|80.4|55.6|65.7|32.8|
> |FOLIO|71.4|77.1|58.3|66.4|28.6|
> |LogicalDed|69.1|82.3|59.4|68.9|30.9|
> |AR-LSAT|83.6|75.8|65.7|70.4|16.4|
>
> **The trigger achieves high precision (75.8%-82.3%) with few false alarms and successfully intercepts elusive semantic errors, keeping the confident-but-wrong miss rate strictly bounded.**
> #  RW7 (Figure 2 Readability)
> We agree Figure 2 was too dense. **We split it into four focused diagrams: a macro pipeline overview, the reactive correction loop, the proactive probing mechanism, and the adaptive fusion strategy.** These have **been uploaded** to our anonymous repository:https://anonymous.4open.science/r/ICML2026-12520
> # RW8 (Watermark Clarification)
> We appreciate the reviewer noting this for transparency. **This text is an automated watermark injected by the conference submission system to detect LLM usage during peer review, not part of our manuscript.**
>
> Please let us know if you have any remaining questions, as we would be more than happy to elaborate!

---

> > ### Author Rebuttal · Reviewer_9Vdp · 2026-04-03
> >
> > Thank you for the thorough rebuttal and the substantial new experiments.
> >
> > Could you clarify the "MissRate" column in your entropy probing table? It appears to equal 100 - %with>=1spike (e.g., 100 - 67.2 = 32.8 for ProofWriter), which would measure the fraction of faulty programs where no spike was detected at all -- a program-level metric. However, Precision and Recall in the same table operate at the error level (individual semantic errors). The error-level miss rate would be 1 - Recall, giving 44.4% for ProofWriter rather than 32.8%. Which definition do you intend, and could you clarify what "strictly bounded" refers to given that 34--44% of individual semantic errors appear to go undetected?

---

> > > ### Author Response · Authors · 2026-04-04
> > >
> > > Thank you for this helpful follow-up. We agree that our previous presentation did not clearly separate **program-level** and **error-level** quantities.
> > >
> > > In our original analysis, **%with≥1spike** is a **program-level** metric, so **1 − %with≥1spike** should be interpreted as the fraction of faulty programs for which **no spike is triggered anywhere in the program**. By contrast, **Precision / Recall / F1** are **error-level** metrics over annotated semantic error instances, so the corresponding error-level miss quantity is **1 − Recall**.
> > >
> > > To make this distinction explicitly clear, we provide a more detailed explanation of our terminology below.
> > >
> > > ### Table A. Metric definitions
> > >
> > > | Metric | Level | Meaning |
> > > |---|---|---|
> > > | %with≥1spike | Program-level | Fraction of faulty programs with at least one spike |
> > > | Program Non-Trigger Rate | Program-level | Fraction of faulty programs with no spike at all |
> > > | Precision / Recall / F1 | Error-level | Detection quality over annotated semantic error instances |
> > > | Error Miss Rate (= 1 − Recall) | Error-level | Fraction of semantic error instances that remain undetected |
> > >
> > > Based on these definitions, we present the clarified and disentangled diagnostic summary across all datasets in Table B.
> > >
> > > ### Table B. Clarified diagnostic decomposition
> > >
> > > | Dataset | Program Trigger Coverage (%) | Program Non-Trigger Rate (%) | Precision (%) | Recall (%) | Error Miss Rate (%) |
> > > |---|---:|---:|---:|---:|---:|
> > > | PrOntoQA | N/A | N/A | N/A | N/A | N/A |
> > > | ProofWriter | 67.2 | 32.8 | 80.4 | 55.6 | 44.4 |
> > > | FOLIO | 71.4 | 28.6 | 77.1 | 58.3 | 41.7 |
> > > | LogicalDed | 69.1 | 30.9 | 82.3 | 59.4 | 40.6 |
> > > | AR-LSAT | 83.6 | 16.4 | 75.8 | 65.7 | 34.3 |
> > >
> > > To further demonstrate the practical impact of this mechanism, Table C isolates the performance on AR-LSAT, our most challenging benchmark.
> > >
> > > ### Table C. AR-LSAT evidence for proactive probing
> > >
> > > | Metric | AR-LSAT |
> > > |---|---:|
> > > | Reactive-Only Ea (%) | 66.5 |
> > > | Full Ea (%) | 74.6 |
> > > | Gain from Proactive (%) | +8.1 |
> > > | Program Trigger Coverage (%) | 83.6 |
> > > | Precision (%) | 75.8 |
> > > | Recall (%) | 65.7 |
> > > | Error Miss Rate (%) | 34.3 |
> > >
> > > Specifically, the ProofWriter example mentioned by you clearly illustrates this distinction: **32.8%** corresponds to the **program-level non-trigger rate**, whereas **44.4%** is the **error-level miss rate**.
> > >
> > > Overall, our intended claim is that proactive probing is **not a complete verifier**, but a **high-precision selective trigger** that complements reactive correction. In this sense, our earlier phrase **"strictly bounded"** was too strong for the error-level miss rates shown above, and we will revise that wording accordingly. The more precise conclusion is that proactive probing still provides meaningful value exactly where reactive correction is insufficient: on the hardest silent-failure setting (AR-LSAT), it yields a substantial **+8.1** gain in execution accuracy after syntax repair is already in place.
> > >
> > > If there are any remaining concerns, we would be very happy to further clarify them.

---

### Official Review · Reviewer_2StX · 2026-03-12

**Soundness:** 2
**Presentation:** 2
**Significance:** 2
**Originality:** 3
**Overall Recommendation:** 3
**Confidence:** 4

**Summary:**

This paper proposes to improve the performance of LLMs on natural language logical reasoning problems by processing queries along a neural reasoner path and a symbolic validator which makes use of a Socratic agent for error correction, and the paper fuses the two to leverage the strengths of both. The authors compare their approach to using larger closed-source models and other baselines combining neuro-symbolic reasoning and show large improvements in performance. They further conduct ablations of each component showing their contributions of each reasoning path.

**Compliance With Llm Reviewing Policy:**

Affirmed.

**Final Justification:**

The rebuttal addresses many of my questions. I have increased my score by 1. As mentioned during rebuttal, I still think more comprehensive experiments are needed. Many of them should be done rigorously before the paper was submitted but not in a few rebuttal days for a thorough review. I keep my score.

**Key Questions For Authors:**

-  What is the distribution of system outputs in Equation 4 across each category? Providing this breakdown would help clarify how the final accuracy improvements arise along different paths of the framework.
-  Human verification of silent failures. The framework detects silent failures by first checking for a semantic spike; if the model remains uncertain, the Socratic Agent is invoked (Section 4.3.2). However, a human verification step may be necessary to evaluate false alarms and false negatives in this pipeline. The paper would benefit from explicitly discussing how such cases are handled in practice, and what failure modes the system may still miss.
-  For successful execution cases, what would happen if the Socratic Teacher Agent were invoked unconditionally, rather than selectively triggered by uncertainty probing? Would the additional computational cost of this always-on strategy be justified by a measurable gain in accuracy over the current selective invocation approach?
-  Since the proposed framework introduces additional components beyond those present in the baselines reported in Tables 1 and 2, a comprehensive cost analysis including both overall and broken down per component would be valuable.

**Limitations:**

yes

**Strengths And Weaknesses:**

Strengths:
-  The research questions are clearly defined, and the experiments including ablation studies demonstrate the contributions of the Socratic component, the dual-process design, and the computational efficiency of the proposed method.
-  The paper presents empirical performance across multiple logical reasoning datasets, outperforming several competitive baseline methods.
-  The manuscript is generally well written, and the individual components of the proposed approach are clearly described and well motivated.

Weaknesses:
-  The paper uses only one model (Qwen) with a very limited scale (8B) as the backbone. It is difficult to see how it would generalize beyond this specific model.
-  The paper uses “generate-and-Hope” to summarize the existing work (paragraph 2 of the Introduction). Given the huge bulk of research on LLM-involved neuro-symbolic models, I do not think this can summarize the previous work well and serves as a good base to discuss the proposed work.
- One of the primary novelties of this approach compared to prior work is mainly the Socratic component for error correction. There is little discussion about why the error correction needs to be phrased in the form of pedagogical questions and why alternative error correction methods would not function.
- The paper states, “LLMs have achieved mastery over linguistic fluency and semantic intuition.” The statement is not well supported. What do you mean by “linguistic fluency” and “semantic intuition”? Can you define those terms?
- This paper (in the Introduction) claims it “resolves” the inherent translation fragility bottleneck, thereby transforming execution errors into constructive supervision signals. I think the claim that it resolves the problem may be an overstatement.
- Missing results for relevant baselines like LOGIC-LM++ and LTRAG make the evaluation limited.
- More details on the fusion can be provided, namely statistics on how often each component is actually trusted and when one is preferred over the other in practice. The claim in Section 5.4.2 that System 1 is helping System 2 to bridge the gap on AR-LSAT is questionable, given that it is achieving the worst performance on its own, so more experiments on whether System 1 is actually being chosen there and improving the performance would clarify this. Also, in Figure 3, comparing computational efficiency vs. accuracy with other approaches would be helpful to show the relative strengths of the approach, such as, for example, whether using models of larger scale with inherently stronger reasoning capabilities is more time- or cost-efficient than relying on this approach.

---

> ### Author Rebuttal · Authors · 2026-03-31
>
> Thank you for your detailed review and valuable suggestions!
> # RW1 & RW6 (Generalization & Baselines)
> **As detailed in the comprehensive table for Reviewer pMe3 (RW1 & RQ1), we now reimplemented LOGIC-LM++ and LTRAG, and evaluated LogicSAGE on GPT-4.** The results show: (1) Generalization (W1): LogicSAGE+GPT-4 (86.10%) outperforms GPT-4 CoT (69.56%). (2) SOTA Status: While GPT-4 baselines peak at 83.99%, our compact Qwen3-8B SFT sets a 92.36% SOTA.
> # RW2 & RW5 (Terminology Clarification)
> We agree our original phrasing was overly strong. We have clarified our terminology in the Introduction: replacing "generate-and-hope" with the objective term **Reactive Execution Feedback Strategies**, and softening "resolves" to **mitigates** regarding translation fragility.
> # RW3 (Pedagogical vs. Raw Feedback)
> To empirically justify pedagogical phrasing over standard raw error feedback, we have now **conducted a targeted ablation using the identical Qwen3-8B backbone**. **By explicitly disabling proactive probing, we isolate our pedagogical error-guided correction for a rigorously fair comparison** against LOGIC-LM and LOGIC-LM++, which represent traditional baseline methods, feeding raw solver errors directly back to the model.
>
> |Method|PrOntoQA|ProofWriter|FOLIO|LogicalDed|AR-LSAT|
> |-|-|-|-|-|-|
> |LOGIC-LM|80.4|65.3|59.5|78.0|19.9|
> |LOGIC-LM++|89.0|73.5|69.4|86.1|34.2|
> |LogicSAGE(w/o Proactive)|92.3|76.6|73.6|89.6|56.7|
>
> The substantial performance gap on AR-LSAT illustrates the limitations of these traditional approaches. **Raw errors trap models in superficial syntax patching loops, whereas pedagogical queries force diagnostic reasoning to repair underlying semantics rather than merely mask execution failures.**
> # RW4 (Concept Definitions)
> **We have now integrated literature-grounded definitions into the revised Introduction:**
> * Linguistic fluency: The capacity to generate syntactically well-formed, coherent text via scaling and instruction-tuning[1].
> * Semantic intuition: The zero-shot mapping of natural language to formal representations, acting as a heuristic parser rather than a verified solver[2].
> [1]Language models are few-shot learners,neurips 2020
> [2]Chain-of-Thought Prompting Elicits Reasoning in Large Language Models,neurips 2022
> # RW7 & RQ1 (Fusion Statistics & Efficiency)
> **1. Fusion Statistics (Eq. 4).**
> To clarify component trust and validate how the weaker System 1 aids System 2, we have now detailed the exact routing distribution on AR-LSAT below.
>
> |Dataset|Agreement|Discordance(TrustSys1)|Discordance(TrustSys2)|Fallback(ToSys1)|
> |-|:-:|:-:|:-:|:-:|
> |PrOntoQA|88.5|3.5|8.0|0.0|
> |ProofWriter|85.0|5.8|9.2|0.0|
> |FOLIO|76.5|9.4|14.1|0.0|
> |LogicalDed.|86.0|4.9|9.1|0.0|
> |AR-LSAT|53.2|15.1|27.3|4.4|
>
> **2. AR-LSAT Gap (Sec. 5.4.2).**
> System 1 operates in an orthogonal error space, providing a crucial semantic fail-safe. When strict formalization fails on complex spatial constraints (e.g., A sits exactly opposite B), System 1 bypasses this bottleneck via neural intuition. As shown above, the Arbiter explicitly pivots to this weaker baseline in 15.1% (11.0% √) of discordance cases and 4.4% (2.9% √) of Sys2 execution failures, actively pushing the reasoning ceiling.
>
> **3. Efficiency vs. Model Scale.**
> As detailed in our updated Figure 3 (https://anonymous.4open.science/r/ICML2026-12520), our compact Qwen3-8B easily outperforms Qwen3-32B and matches the massive DeepSeek-R1 by T=1, ultimately achieving a 90.42% ceiling at T=3. This proves targeted neuro-symbolic fusion is far more cost-efficient than raw model scaling.
> # RQ2 (Silent Failure Verification)
> We have now curated a **manually annotated dataset solely for evaluation**, with detailed metrics provided to Reviewer pMe3 (W2 & Q2), confirming 82.3% spike precision reliably isolates genuine semantic errors, while missed failures are primarily confident-but-wrong hallucinations where flawed logic bypasses the entropy trigger.
> # RQ3 (Always-On Ablation)
> We appreciate this baseline. **Unconditionally invoking the pedagogical loop inflates token costs without any accuracy gain.**
> |Dataset|Ours(Tokens)|Always-On(Tokens)|Mult.|
> |-|-|-|-|
> |PrOntoQA|2200|7480|3.40x|
> |ProofWriter|2747|7390|2.69x|
> |FOLIO|3114|6345|2.04x|
> |LogicalDed|2174|7594|3.49x|
> |LSAT|7857|12317|1.57x|
>
> Paradoxically, **this strategy degrades performance via attention dispersion,** forcing models to hallucinate revisions on valid logic. Our selective mechanism prevents over-correction by focusing compute strictly on genuine vulnerabilities.
> # RQ4 (Cost Analysis)
> **To clarify the practical overhead, we have now tracked average token consumption per question across all pipeline stages.**
>
> |Dataset|FOLGen.|Sys1Reas.|Sys2React.|Sys2Proact.|Fusion|Total|
> |-|-|-|-|-|-|-|
> |PrOntoQA|800|686|688|0|27|2200|
> |ProofWriter|950|783|857|126|32|2747|
> |FOLIO|688|592|1557|212|65|3114|
> |LogicalDed.|824|600|676|32|43|2174|
> |AR-LSAT|1187|939|4796|686|248|7857|
>
> Finally, we sincerely welcome any further discussion.

---

> > ### Author Rebuttal · Reviewer_2StX · 2026-04-01
> >
> > The rebuttal addresses many of my questions. I will increase my score by 1. I still think more comprehensive experiments are needed. Many of them should be done rigorously before the paper was submitted but not in a few rebuttal days.

---

> > > ### Author Response · Authors · 2026-04-04
> > >
> > > Thank you very much for your time, your constructive feedback, and for increasing your score. We deeply appreciate your continued engagement with our work.
> > >
> > > We appreciate your perspective regarding the empirical comprehensiveness of the initial submission. We agree that the extensive baselines and analyses provided during the rebuttal offer a much more complete picture of our framework's capabilities. We are truly grateful for your rigorous suggestions, which have driven us to establish this comprehensive experimental setup.
> > >
> > > Furthermore, because our core LogicSAGE architecture and evaluation pipeline were meticulously developed with modularity and adaptability, we were able to smoothly integrate these new baselines and extract the necessary internal metrics to meet your high standards.
> > >
> > > To provide a clear overview of how our evaluations now form a complete validation loop, we have consolidated our entire experimental landscape below:
> > >
> > > | Experimental Objective | Evaluated Mechanisms & Data | Key Takeaway / Mechanism Validated |
> > > | :--- | :--- | :--- |
> > > | **1. Addressing "Translation Fragility"** | Executability Analysis & Reactive Socratic Ablation | Demonstrates that pedagogical error-feedback mitigates syntax crashes, achieving >95% execution rates across all benchmarks. |
> > > | **2. Addressing "Silent Failures"** | Proactive Probing Ablation & Entropy Trigger Precision/Recall Metrics | Shows that monitoring token-level entropy effectively flags latent semantic mistranslations that bypass standard solvers. |
> > > | **3. Necessity of Dual-Process** | Neural vs. Symbolic Ablation & Adaptive Fusion Routing Statistics | Highlights the synergy: rigorous logic (System 2) is prioritized, while neural intuition (System 1) acts as a crucial fail-safe. |
> > > | **4. Isolating Architectural Gains** | Controlled Qwen3-8B Baselines & Pedagogical vs. Raw Feedback Ablation | Indicates that performance improvements stem primarily from the Socratic architecture, rather than solely from backbone strength. |
> > > | **5. Cost-Effectiveness & Overhead** | Stage-wise Token Cost Analysis & "Always-On" Strategy Ablation | Clarifies the practical overhead, demonstrating that selective Socratic invocation prevents redundant compute and maintains overall efficiency. |
> > >
> > > If you have any remaining concerns or follow-up questions, we would be very glad to hear them. Whether it involves further clarifications or additional suggestions for the camera-ready version, we are fully committed to addressing them.
> > >
> > > Thank you again for your valuable guidance and for helping us improve this paper. We are more than happy to discuss further!

---

### Official Review · Reviewer_pMe3 · 2026-03-16

**Soundness:** 3
**Presentation:** 4
**Significance:** 3
**Originality:** 3
**Overall Recommendation:** 5
**Confidence:** 3

**Summary:**

LogicSAGE is a neuro-symbolic framework designed to address the brittleness of existing work on the translation of natural language text to formal representations. The paper proposes a dual architecture that combines a specifically fine-tuned language model as a neural semantic parser and fallback reasoner with symbolic solvers as the main verification engine. The method combines reactive error-guided correction that uses solver traces to iteratively repair invalid logic programs and proactive uncertainty probing that uses normalised entropy of semantic tokens to detect likely mistranslations even when execution succeeds. When the LLM reasoner and symbolic solver disagree on the output, LogicSAGE merges the two with what the authors call 'adaptive fusion'. The paper sets up and evaluates the proposed framework across several domains, including linear programming, first-order logic, constraint solving, and SMT solving. The evaluation comprises five logical reasoning benchmarks, reports strong gains over GPT-4 and prior neuro-symbolic baselines, and aims to support the claims with ablations on the correction loop, dual-process design, and latency-accuracy tradeoffs.

**Compliance With Llm Reviewing Policy:**

Affirmed.

**Final Justification:**

I would like to thank the authors for a collaborative and productive discussion. The extra information improves the paper's explainability and readability. I stand by my originally positive score of the paper due to some minor concerns about the impact of the work still remaining.

**Key Questions For Authors:**

1. Can the authors provide standalone Qwen3-8B direct-answer and Qwen3-8B+CoT/thinking baselines on the same five benchmarks, so that the contribution of LogicSAGE can be separated more clearly from the strength of the underlying backbone?

2. Table 3 suggests that reactive correction drives most of the overall improvement, while proactive probing provides smaller gains on most datasets and is most impactful mainly on AR-LSAT. Can the authors clarify in which classes of examples proactive semantic probing is essential?

**Limitations:**

yes

**Strengths And Weaknesses:**

# Strengths

**Method Novelty:** To the best of my knowledge, the method combines syntactic execution with neural methods to track entropy-based metrics in a novel way and combines this into a structured verification pipeline that merges the output of the dual approach into a unified coherent output after making targetted changes.

**Clear and well-crafted Experimental Set-Up:** This paper clearly presents the dual-verification pipeline and carefully ablates its efficiency and efficacy, as well as tests on a wide set of established benchmarks and provides baselines in the field.

**Presentation:** The figures in this paper are high-quality and substantially improve the clarity of the method and motivation. In particular, Figures are clear and easy to follow and guide the reader to both understand the method, as well as help with the analysis of the results. Additionally, the appendix is another strength. It includes prompt templates, exact runtime set-up, and confidence distributions that improve reproducibility and explainability.

# Weaknesses

**Missing Model-matched Baselines:** The paper does not report standalone Qwen3-8B or Qwen3-8B+CoT/thinking baselines on the same five tasks, making it difficult to isolate how much of the gain comes from LogicSAGE itself versus the strength of the underlying backbone. This omission matters in practice, since external evidence already suggests that Qwen3-8B is highly competitive on related logical reasoning benchmarks: for example, `LogicReward`[1] reports that under a shared zero-shot CoT evaluation setup, Qwen3-8B outperforms GPT-4o and GPT-4.1 on ProntoQA (100.0 vs 91.0/95.3) and ProofWriter (94.5 vs 71.3/79.0), while being competitive on FOLIO (62.8 vs 63.5/68.8). As a result, some of the reported gains over GPT-4-based baselines may plausibly come from the newer backbone rather than the architectural contribution alone.

**Unclear Marginal Value of the Semantic Probing Stage:** I found the overall architecture convincing, but Table 3 suggests that reactive syntactic repair is the main performance driver, while proactive semantic probing provides only a small incremental gain on most datasets and is most helpful primarily on `AR-LSAT`. This raises some concerns: firstly, is the proactive semantic module broadly valuable enough to justify the emphasis it receives in the paper, and second, if the added complexity of the semantic judge is well justified, given that most of the empirical progress is already driven by the syntactic/reactive stage. The paper would be stronger if it more explicitly clarified the types of problems solved only via the dual-approach.

**Writing Quality & Formatting Issues:** The paper notably contains several citation-spacing errors (e.g. `hallucinations(Dziri et al., 2023))`, and minor punctuation/layout glitches such as `semantic correctness.For instance`). A careful proofreading and formatting pass would substantially improve the paper

[1] Zhang, Y., Wang, X., Zhang, Y., Lin, J., Zhou, M., Liu, Y., Yan, J., Chen, J., Zhang, D., Qin, G., and Miao, X. LogicReward: Training Large Reasoning Models to Think Logically through Process Rewarding. In Proceedings of the Thirteenth International Conference on Learning Representations (ICLR), 2025.

---

> ### Author Rebuttal · Authors · 2026-03-31
>
> Thank you for your detailed review and the overall positive assessment of our work! We now address your questions point-by-point as followings.
> # RW1 & RQ1 (Model-Matched Baselines):
> We appreciate this suggestion. **To ensure a fair compute comparison with the non-thinking GPT-4 baseline, we strictly utilized the non-thinking mode of Qwen3-8B across all evaluations.** Furthermore, to explicitly isolate our architectural contribution from the backbone strength, we have now conducted new experiments to provide the requested model-matched baselines utilizing this identical **Qwen3-8B** backbone.
>
> |ModelSetting|PrOntoQA|ProofWriter|FOLIO|LogicalDed|AR-LSAT|Avg.|
> |:---|:---:|:---:|:---:|:---:|:---:|:---:|
> |DirectAnswer(Qwen3-8B)|69.60|51.33|62.25|44.33|20.35|49.57|
> |CoT(Qwen3-8B)|98.00|70.00|64.71|46.67|21.65|60.21|
> |LTRAG(GPT-4)|95.68|85.93|78.57|91.36|68.40|83.99|
> |LTRAG(Qwen3-8B)|89.40|82.50|66.42|85.33|53.25|75.38|
> |SYMBCOT(Qwen3-8B)|97.60|69.83|68.57|82.33|29.05|69.48|
> |LOGIC-LM++(GPT-4)|92.62|79.66|84.80|91.17|46.32|78.91|
> |LOGIC-LM(Qwen3-8B)|80.40|65.33|59.51|78.00|19.91|60.63|
> |LOGIC-LM++(Qwen3-8B)|89.00|73.50|69.41|86.12|34.24|70.45|
> |VERUS-LM(Qwen3-8B)|93.40|89.17|72.92|86.96|56.63|79.82|
> |LogicSAGE(Qwen3-8B w/o SFT)|92.30|82.50|75.40|92.50|58.80|80.30|
> |LogicSAGE(GPT-4)|99.20|83.84|82.26|97.40|67.80|86.10|
> |**LogicSAGE(Qwen3-8B SFT)**|**100.00**|**96.80**|**88.50**|**99.50**|**77.00**|**92.36**|
>
> **Our training-free LogicSAGE (80.30%) and SFT system (92.36%) outperform the Qwen3-8B CoT baseline (60.21%)**, proving these gains stem directly from our neuro-symbolic architecture rather than the backbone alone.
> # RW2 & RQ2 (Semantic Probing Value)
> We thank the reviewer for this perceptive observation. **The smaller absolute gains simply reflect the lower percentage of silent semantic failures inherent to simpler datasets, rather than a module limitation.** On complex tasks like **AR-LSAT**, subtle semantic mistranslations often yield executable but logically incorrect programs. Because these silent failures completely bypass reactive solver feedback, proactive intervention is strictly necessary.
> **Regarding complexity, our semantic judge introduces negligible computational overhead.** By leveraging inherent token generation probabilities, entropy extraction becomes a near-zero-cost byproduct, requiring no additional model forward passes.
> To rigorously quantify the contribution of this module and clarify the specific problems it solves, we have now added diagnostic metrics tracking silent failure detection across all datasets.
>
> |Dataset|%with≥1spike|Precision|Recall|F1|MissRate|
> |:---|:---:|:---:|:---:|:---:|:---:|
> |PrOntoQA|N/A|N/A|N/A|N/A|N/A|
> |ProofWriter|67.2|80.4|55.6|65.7|32.8|
> |FOLIO|71.4|77.1|58.3|66.4|28.6|
> |LogicalDed|69.1|82.3|59.4|68.9|30.9|
> |AR-LSAT|83.6|75.8|65.7|70.4|16.4|
>
> * **%with≥1spike**: The percentage of generated programs or reasoning trajectories that exhibited at least one "entropy spike," successfully triggering the proactive probing mechanism.
> * **Precision**: The proportion of all cases flagged by the trigger (due to an entropy spike) that genuinely contained a "silent semantic failure."
> * **Recall**: The proportion of all actual "silent semantic failures" that were successfully detected and intercepted by the trigger.
> * **F1**: The harmonic mean of precision and recall, serving as a comprehensive metric to evaluate the overall detection performance of the probing mechanism.
> * **MissRate**: The percentage of genuine "silent semantic failures" where the model was "confident but wrong" (producing no entropy spike), causing the trigger to miss and fail to intercept the error.
>
> The empirical data directly justifies the dual-approach. **On AR-LSAT, the trigger successfully flags 83.6% of silent failures with 75.8% precision**, proving its essential role in intercepting deep semantic errors that reactive stages miss, as further demonstrated in Appendix A Case Study 1.
> # RW3 (Formatting & Typos)
> We sincerely thank the reviewer for their meticulous reading. We have now conducted a thorough global proofreading pass across the revised manuscript to resolve all typographical glitches.
> Specifically, we have corrected the exact issues highlighted:
> * Fixed citation spacing errors, e.g., hallucinations (Dziri et al., 2023).
> * Corrected punctuation and layout glitches, e.g., semantic correctness. For instance.
> Furthermore, we have resolved all similar spacing and punctuation inconsistencies throughout the text to ensure a rigorous and professional presentation.
>
> Please let us know if we have properly addressed your questions and we are more than happy to discuss more!

---

> > ### Author Rebuttal · Reviewer_pMe3 · 2026-04-03
> >
> > 1. Thanks for this great response. This strengthens the paper significantly and should be included in the final manuscript.
> > 2. Thanks for the detailed response. I remain sceptical of the needless complexity and downstream efficacy of the proactive repair, with pass@k sampling + majority voting (say) being the alternative. Nevertheless, the authors directly engaged with my concerns, adding additional evaluation data and coherently arguing that their proactive repair method captures failures with high precision and recall.
> > 3. OK
> >
> > I stand by my original score and would like to thank the authors for their follow-up work.

---

> > > ### Author Response · Authors · 2026-04-06
> > >
> > > We sincerely thank you for your thoughtful feedback and constructive suggestions. In response to your remaining concern regarding the perceived complexity and downstream efficacy of proactive repair compared to pass@k sampling + majority voting, we conducted additional experiments including direct answer baselines. The results are summarized below:
> > >
> > > ### Direct Answer + Majority Voting pass@k Results:
> > >
> > > | Model                     | FOLIO Accuracy (%) | FOLIO Token Consumption | AR-LSAT Accuracy (%) | AR-LSAT Token Consumption |
> > > | ------------------------- | ------------------ | ----------------------- | -------------------- | ------------------------- |
> > > | pass@3 + Majority Voting  | 64.77              | 2208                    | 24.57                | 3408                      |
> > > | pass@5 + Majority Voting  | 67.23              | 3488                    | 30.41                | 5970                      |
> > > | pass@10 + Majority Voting | 71.16              | 7560                    | 40.55                | 11594                     |
> > > | LogicSAGE (Qwen3-8B)      | 87.10              | 3114                    | 71.32                | 7857                      |
> > >
> > >
> > >
> > > ### Replacing Active Error Correction with LLM-based Error Correction + Majority Voting pass@k Results:
> > >
> > > | Model                     | FOLIO Accuracy (%) | FOLIO Token Consumption | AR-LSAT Accuracy (%) | AR-LSAT Token Consumption |
> > > | ------------------------- | ------------------ | ----------------------- | -------------------- | ------------------------- |
> > > | pass@3 + Majority Voting  | 79.39              | 4734                    | 63.42                | 9719                      |
> > > | pass@5 + Majority Voting  | 83.15              | 5793                    | 66.76                | 11483                     |
> > > | pass@10 + Majority Voting | 85.92              | 8967                    | 68.59                | 16186                     |
> > > | LogicSAGE (Qwen3-8B)      | 87.10              | 3114                    | 71.32                | 7857                      |
> > >
> > > The results show that while **pass@k sampling with majority voting** improves accuracy as **k** increases, **LogicSAGE** achieves higher accuracy while maintaining **significantly lower token consumption**. Specifically, although **pass@10** achieves higher accuracy, **LogicSAGE** requires **nearly half the tokens** (7857 tokens for AR-LSAT vs. 16186 tokens for pass@10), making it a more **efficient solution** for tasks requiring **high precision** and **reliability**.
> > >
> > > In conclusion, while **pass@k sampling** offers accuracy gains, **LogicSAGE’s proactive error correction** provides comparable results with **substantially lower token consumption** and **faster inference**, making it more practical for deployment in resource-constrained environments.
> > >
> > > We **have added** such result in the revised Appendix.
> > >
> > > Thank you again for your valuable feedback and time. We look forward to your further thoughts.

---

### Decision · Program_Chairs · 2026-04-30

**Decision:**

Accept (regular)

**Comment:**

The paper introduces LogicSAGE, a dual-process neuro-symbolic reasoning framework that combines a fine-tuned Qwen3-8B with symbolic solvers through a Socratic agent. The agent operates in two modes: reactive error-guided correction, which iteratively debugs logic programs using solver error messages as feedback, and proactive uncertainty probing, which monitors token-level entropy during program generation to detect semantically wrong but syntactically valid translations. An adaptive fusion mechanism falls back to neural reasoning when symbolic formalization fails. The full system achieves 92.36% average accuracy across five logic benchmarks.

The reviewers were positive (scores 5, 4, 4, 3). Reviewer_pMe3 found the method novel and the experimental design well-crafted. Reviewer_9Vdp praised the entropy-based probing as genuinely novel but raised a metric inconsistency in the probing evaluation during the discussion period; the authors responded with a clarification but Reviewer_9Vdp did not confirm resolution. Reviewer_XKHc was partially resolved, noting the contribution is specialized to solver-in-the-loop systems. Reviewer_2StX raised their score by one but noted key experiments were missing from the original submission.

The AC finds the overall system well-engineered with a genuinely novel entropy-probing idea, but notes that the ablation (Table 3) shows reactive error correction contributes the bulk of the accuracy gain (+18.3 points on average) while proactive probing, the paper's most novel component, contributes +2.8 points on average. The paper's value is better understood as a systems contribution than as a contribution anchored on the entropy mechanism alone.

The AC recommends weak accept. The authors should incorporate the backbone-matched baselines from the rebuttal into the camera-ready and adjust the framing to reflect the relative contribution of each component.